# Gene-selective transcription promotes the inhibition of tissue reparative macrophages by TNF

Stefanie Dichtl[1] , David E Sanin[2,9], Carolin K Koss[3], Sebastian Willenborg[4], Andreas Petzold[5], Maria C Tanzer[1], Andreas Dahl[5] , Agnieszka M Kabat[2,9] , Laura Lindenthal[1], Leonie Zeitler[1] , Sabrina Satzinger[4], Alexander Strasser[1], Matthias Mann[1] , Axel Roers[6], Sabine A Eming[4,7,8,10], Karim C El Kasmi[3], Edward J Pearce[2,9], Peter J Murray[1]

**Anti-TNF therapies are a core anti-inflammatory approach for chronic diseases such as rheumatoid arthritis and Crohn's Disease. Previously, we and others found that TNF blocks the emergence and function of alternative-activated or M2 macrophages involved in wound healing and tissue-reparative functions. Conceivably, anti-TNF drugs could mediate their protective effects in part by an altered balance of macrophage activity. To understand the mechanistic basis of how TNF regulates tissue-reparative macrophages, we used RNAseq, scRNAseq, ATACseq, time-resolved phospho-proteomics, gene-specific approaches, metabolic analysis, and signaling pathway deconvolution. We found that TNF controls tissue-reparative macrophage gene expression in a highly gene-specific way, dependent on JNK signaling via the type 1 TNF receptor on specific populations of alternative-activated macrophages. We further determined that JNK signaling has a profound and broad effect on activated macrophage gene expression. Our findings suggest that TNF's anti-M2 effects evolved to specifically modulate components of tissue and reparative M2 macrophages and TNF is therefore a context-specific modulator of M2 macrophages rather than a pan-M2 inhibitor.**

## Introduction

A temporal and spatial balance between pro- and anti-inflammatory immune responses ultimately defines the final outcome of infection, wounding, and tissue repair. In general, the passage from an inflammatory response associated with microbial clearance, cellular damage, and cell death that evolves towards tissue repair and homeostasis is termed resolving inflammation (Murray & Wynn, 2011;

Murray, 2017). By contrast, nonresolving inflammation underlies chronic diseases, inability to clear infections, many autoimmune syndromes and severe organ and tissue damage from mechanical injury and burns (Nathan & Ding, 2010; Eming et al, 2017). As cytokines have a central role in establishing the balance between pro- and anti-inflammatory immunity, their manipulation in chronic disease has been, and remains, a core interest for development of therapeutic interventions that seek to mitigate the consequences of non-resolving inflammation.

TNF is a pro-inflammatory cytokine targeted by different biologic drugs including Infliximab, a monoclonal antibody directed against both soluble and transmembrane bound forms of TNF, and Etanercept, an engineered soluble TNF receptor (Youngquist et al, 2013). The rationale for using anti-TNF drugs in rheumatoid arthritis (RA), inflammatory bowel disease (IBD) and psoriasis centered on clinical and experimental data linking pro-inflammatory activities of TNF to non-resolving inflammation. A central finding that promoted the concept that anti-TNF drugs could be effective in RA came from the ex vivo evaluation of the inflammatory milieu in synovial cultures from RA patients. In this system, TNF blockade neutralized TNF but also IL1$\beta$ leading Feldmann and colleagues to suggest TNF was at the apex of a pro-inflammatory cascade and that its inhibition could break a non-productive inflammatory cycle (Brennan et al, 1989; Feldmann, 2002). In other words, inhibiting TNF reduces inflammation by interrupting non-resolving inflammation. Anti-TNF drugs were thereafter used in millions of patients with chronic inflammation.

Recently, an additional and unexpected connection between TNF blockade and resolving inflammation was exposed using molecular genetic approaches. In mice lacking the type 1 tumor necrosis factor receptor (TNFR1, encoded by Tnfr1a) or TNF (encoded by Tnf), diverse inflammatory responses were associated with a striking enhancement of macrophages with M2 characteristics (Kroner et al, 2014; Kratochvill et al, 2015b; Schleicher et al, 2016; Li et al, 2017,

[1]Max Planck Institute of Biochemistry, Martinsried, Germany   [2]Department of Immunometabolism, Max Planck Institute for Immunobiology and Epigenetics, Freiburg, Germany   [3]Boehringer Ingelheim Pharma GmbH and Co KG, Biberach, Germany   [4]Department of Dermatology, University of Cologne, Cologne, Germany   [5]Deep Sequencing Group, Biotechnology Center, Technische Universität Dresden, Dresden, Germany   [6]Institute for Immunology, Medical Faculty Carl Gustav Carus, TU Dresden, Dresden, Germany   [7]Center for Molecular Medicine Cologne, University of Cologne, Cologne, Germany   [8]Cologne Excellence Cluster on Cellular Stress Responses in Aging-Associated Diseases, University of Cologne, Cologne, Germany   [9]The Bloomberg~Kimmel Institute for Cancer Immunotherapy at Johns Hopkins, Johns Hopkins University, Baltimore, MD, USA   [10]Institute of Zoology, Developmental Biology Unit, University of Cologne, Cologne, Germany

Correspondence: murray@biochem.mpg.de

2021). M2 macrophages (also referred to as alternatively activated macrophages) are enriched in inflammation associated with type 2 immune responses, IL4 and IL13 production, and transitions to wound healing and tissue repair. M2 macrophages have a hallmark gene expression profile consisting of numerous mRNAs encoding proteins that control T cell immunosuppression (Arginase-1 or Arg1), matrix remodeling (Fibronectin or Fn1) and a unique pattern of secreted proteins (Retnla or Relma/Fizz1, Chil3 or Ym1, a chitinase) and cell surface markers (Clec10a or CD301/Mgl, a cell surface receptor, Mgl2, a lectin and Mrc1 or CD206 or the mannose receptor) (Murray, 2017). Many M2 genes are regulated by transcription factors including STAT6 and PPARδ/γ (Pauleau et al, 2004; Vats et al, 2006; Odegaard et al, 2007), which, in addition to controlling wound healing and tissue repair gene expression, also sustain mitochondrial oxidative phosphorylation (OXPHOS) necessary for fatty acid oxidation, a key metabolic feature of M2 macrophages (Vats et al, 2006; van Teijlingen Bakker & Pearce, 2020; Eming et al, 2021; Kieler et al, 2021).

An implication of the discovery of the anti-M2 effect of TNF is that in clinical settings where suppression of the pro-inflammatory effects of TNF is the intended mechanism-of-action, M2 macrophages will be enhanced in number or function (Vos et al, 2011, 2012). Thus, an increase in the local proportion of M2 macrophages could aid in tissue repair and a return to a balanced inflammatory setting, at least in patients responsive to the drugs. Such an effect would occur simultaneously with the reduction in TNF's pro-inflammatory effects. However, the molecular mechanism(s) of TNF-mediated control of M2 macrophages remains unresolved.

Signaling from TNFR1 involves three main pathways: NF-κB signaling, MAPK signaling and the caspase-8 pathway, which intersects with the receptor-interacting serine/threonine-protein kinase 3 (RIPK3) and RIPK1 pathways to regulate necroptosis and apoptosis (Jang et al, 2021). Activation of the AP-1 family of basic leucine zipper (bZIP) transcription factors is a major target of TNF-activated MAPK signaling (Karin, 1995). Conceivably, any one of these pathways alone or in combination could suppress M2 macrophage gene expression by different mechanisms. Using scRNAseq, RNAseq, ATACseq, global phosphoproteomics, and gene-specific mechanistic studies, we found that whereas TNF regulates many hundreds of genes in macrophages, its effects on IL4-regulated gene expression are highly gene specific. TNF bypasses the IL4 activation of STAT6 to control key M2-associated genes, while leaving hundreds of other IL4-regulated genes unaffected and having a negligible effect of the OXPHOS metabolic phenotype of M2 macrophages. We traced the anti-TNF signaling pathway to JNK signaling, which has a profound and broad effect on macrophage gene expression. Our findings argue that TNF's anti-M2 effects evolved to specifically modulate components of tissue and reparative M2 macrophages.

## Results

### TNF inhibits M2 gene expression

When macrophages are stimulated with IL4 (with or without IL13, which also signals through the IL4Rα; hereafter we used a combination of IL4+IL13 to harmonize the results with previous studies [Kratochvill et al, 2015b]), a characteristic gene expression program

is activated, which was first described as "alternative activation" (Gordon & Martinez, 2010) and often referred to as "M2." The physiology of only a handful of M2-associated targets, such as Arg1 and PD-L2, are understood in terms of their relationship to M2 functions in type 2 immunity and metabolic maintenance of OXPHOS, which is essential for the M2 phenotype (Pesce et al, 2009; Huber et al, 2010; Gundra et al, 2017; Van de Velde et al, 2017; Baardman et al, 2018; Wang et al, 2018; Tavukcuoglu et al, 2020). To begin to dissect the influence of TNF on M2 gene expression in molecular terms, we chose one M2 target, *Retnla* (encoding RELMα), as a paradigmatic gene induced by IL4+IL13 and negatively regulated by TNF (Kratochvill et al, 2015b). When primary BMDMs were stimulated with IL4+IL13 over a 24-h period, *Retnla* expression increased in two phases, beginning at ~6 h and then increasing to ~5,000-fold that of control unstimulated macrophages by 24 h (Fig 1A). When TNF was added with IL4+IL13, expression of the *Retnla* mRNA was completely inhibited. This was independent of TNF-mediated effects on the temporal activation of pSTAT6 by IL4+IL13 (Fig 1B), indicating that TNF does not interfere in the key first step in IL4Rα receptor activation, necessary for the entire M2 pathway.

Consistent with the effect of TNF on the *Retnla* mRNA, TNF blocked RELMα protein expression (Figs 1C and S1A and B) and this effect was partly reversed by Etanercept (Figs 1D and S1D), which reduces the effects of TNF but does not completely eliminate the function of the cytokine (Cai et al, 2020). Only ~20% of the cells stimulated with IL4+IL13 were positive for RELMα, highlighting the heterogeneity of the M2-associated protein expression in BMDMs. Similarly, TNF inhibited M2 targets CD301 and Arg1 on protein level (Fig S1B–E). In vivo, we observed a similar effect of TNF blockade: in the skin wound healing model, which follows a predictable kinetic from early inflammation to wound repair and scarring (Knipper et al, 2015; Hesketh et al, 2017), *Retnla* mRNA expression increased in the late stage tissue repair phase and was enhanced when mice were treated with Etanercept (Figs 1E and S1F). Furthermore, small peritoneal macrophages, which are derived from the monocyte pool, express M2 markers such as Arg1 and RELMα (Gundra et al, 2017; Krljanac et al, 2019). Therefore, we injected Etanercept or solvent intraperitonally and analyzed the small peritoneal macrophages. We were able to show that solvent treated small peritoneal macrophages express a high Arg1 expression compared with Etanercept treated ones where this pattern was shifted in favor of RELMα expression highlighting the different expression of M2 targets in vivo (Figs 1F and S1G). To furtherpoint out that other well-known M2 genes behave comparable to *Retnla*, we detected impaired CD206 (encoded by Mrc1) expression of IL4+IL13+TNF stimulated BMDMs compared with IL4+IL13–treated ones (Fig S2A). The analysis of the in vivo wound model showed a similar trend towards higher *Mrc1* expression in Etanercept-injected mice in the late stage tissue repair phase (Fig S2B) and SPMs from Etanercept treated mice had an increased expression of CD206 (Fig S2C). Based on our in vitro and in vivo experiments, *Retnla* was therefore used as a "standard" gene strongly induced by IL4+IL13 and inhibited by TNF.

### TNF selectively blocks M2 gene expression

As a first step to understanding how TNF regulates M2 gene expression, we considered two broad possibilities. Having excluded

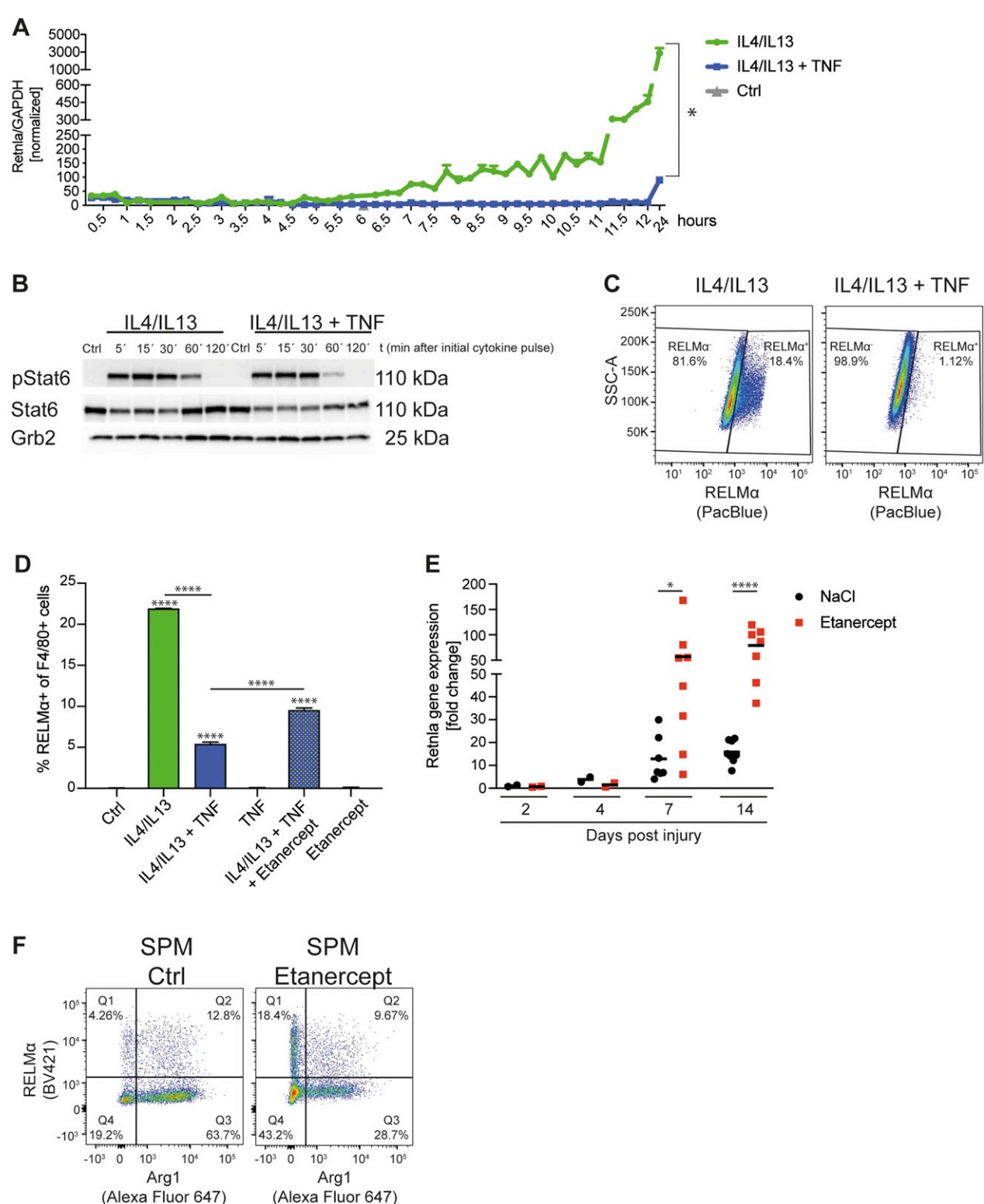

**Figure 1. TNF inhibits M2 gene expression.**
**(A)** RNA from wild-type BMDMs left untreated (Ctrl), stimulated with IL4/IL13 or IL4/IL13 + TNF was used for qRT-PCR. Data shown are the mean fold-increase compared with the Ctrl group. **(B)** Wild-type BMDMs, which were treated with solvent (Ctrl), IL4/IL13 or IL4/IL13 + TNF for 5, 15, 30, 60, or 120 min, were used to isolate whole-cell lysates and analyzed by Western blotting for the indicated proteins. **(C)** Representative plots of wild-type BMDMs stimulated with IL4/IL13 or IL4/IL13 + TNF and percentages of RELMα⁺ of F4/80⁺ macrophages were determined by flow cytometry analysis. **(D)** Wild-type BMDMs were stimulated with IL4/IL13, IL4/IL13 + TNF and/or Etanercept and percentages of RELMα⁺ macrophages were determined by flow cytometry analysis. Data represent three independent experiments. **(E)** CD45⁺CD11b⁺F4/80⁺ macrophages from skin excisional wounds were sorted, RNA was extracted and analyzed by qRT-PCR. Data are representative for two independent experiments. **(F)** Small peritoneal

obvious effects of TNF on IL4Rα-mediated activation of STAT6 (Fig 1B) we thought that TNF could negatively regulate a common event in IL4Rα signaling, or PPARγ or IRF4, key transcription factors required for M2 gene expression. In this case, TNF would block all or most IL4Rα-mediated gene expression. In the second possibility, TNF would regulate specific M2 targets. Conceptually, a corollary to the latter pathway is IL10 anti-inflammatory signaling, which is highly gene-selective for inhibition of key TLR-activated genes rather than acting as a general blocker of TLR gene expression (Lang et al, 2002; Murray, 2005; Murray & Smale, 2012). To distinguish between these possibilities, we used RNAseq to measure TNF-regulated gene expression in BMDMs co-stimulated with IL4+IL13 for 6 or 24 h (Fig 2A), correlated to the biphasic expression of the *Retnla* mRNA (Fig 1A). The results of these experiments demonstrated TNF selectively rather than generally inhibited M2 gene expression. At 6 or 24 h > 800 transcripts were increased above baseline by IL4+IL13; of these, only 20 or 13, respectively were blocked by the combination of IL4+IL13 and TNF and included the *Retnla* mRNA (Fig S3A and B). TNF, by comparison, up-regulated >500 transcripts and most TNF-induced mRNAs were not or only minimal up-regulated by the combination of IL4+IL13+TNF (Fig S3C). Furthermore, TNF also acted with IL4+IL13 to increase the expression of mRNAs including those encoding IRF4, PD-L2 (encoded by Pdcd1lg2), and CCL8 (Fig 2B).

By examining the patterns of the mRNAs inhibited by TNF, we identified three broad modes of regulation (Fig 2C and D). Type 1 mRNAs were increased with IL4+IL13 stimulation from baseline and the addition of TNF blocked this increase (e.g., *Retnla*). Type 2 mRNAs were also increased with IL4+IL13 but decreased from baseline with the supplementation of TNF (e.g., *Mertk*). Type 3 mRNAs were not increased by IL4+IL13 but diminished by TNF at baseline (e.g., *Cd5l*) and therefore are not conventional M2 genes, as they are not induced by IL4+IL13. We verified these regulatory modalities by qRT-PCR (Fig S3D). One striking aspect of the overall negative effect of TNF is that many of the targeted mRNAs are implicated in hallmark M2 tissue repair and resolution functions such as matrix remodeling (*Retnla*, *Fn1*), immune regulation (*Arg1*, *Ccl2*), and phagocytosis (*Mrc1* also known as CD206 or the mannose receptor); this pattern is consistent with the original observations about TNF restricting M2 macrophages, as TNF targets key mRNAs needed for the execution of tissue repair and wound healing (Kratochvill et al, 2015b; Schleicher et al, 2016; Li et al, 2017). Nevertheless, the inhibitory effects of TNF on the IL4+IL13 transcriptome was remarkably selective given the substantial numbers of genes regulated by IL4+IL13, TNF, or the combination of IL4+IL13+TNF.

### Heterogeneity in TNF-responsive macrophages resolved at single cell resolution

Given the selective effect of TNF towards M2 target genes and the heterogeneity of responders during anti-TNF therapy (Coenen et al, 2007; Schett et al, 2021), we next wondered how this pattern would manifest at the single cell level. In other words, did the effect of TNF

observed in bulk RNAseq obscure a higher granularity of population-level signaling? We compared IL4+IL13-stimulated macrophages (24 h) with or without TNF stimulation using the SmartSeq scRNAseq platform to capture transcript information from 384 cells in each condition (in two independent experiments), which gives greater sequence coverage compared to 10X scRNAseq (Picelli et al, 2013; See et al, 2018). Consistent with recent scRNAseq experiments showing heterogeneity of bone-marrow sourced CSF1-dependent macrophages (Muñoz-Rojas et al, 2021), BMDMs treated with IL4+IL13 showed nine distinct sub-populations (Fig 3A). The 10 most regulated genes from each cluster are displayed in Fig S4A. When we compared the IL4+IL13 pattern with addition of TNF, cluster 1 and 2 disappeared and was replaced with a new cluster 0 (Fig 3B). Minor cluster 6 was also reduced with the co-stimulation of TNF. Clusters 5 and 8 were increased in response to TNF stimulation (Fig S4B). The other minor populations within the culture were not obviously affected by TNF.

At first glance, the differential effect of TNF on populations 1 and 2 could reflect differential IL4, IL13, or TNF receptor expression on each subpopulation. However, mRNAs encoding *IL4rα*, *IL13Rα*, and both TNFRs were detected in all clusters equally (Fig S4C). The effect of TNF on the expression of M2 targets differed depending on the macrophage population and target gene. For example, expression of *Clec10a* was extinguished in all populations consistent with our qRT-PCR and bulk RNAseq observation, whereas the *Mgl2* mRNA was reduced in populations 1 and 2, for example, which are enriched for most of the M2 "signature" mRNAs (Figs 3C and S4D). *Retnla*, *Clec10a*, *Mgl2*, *Mrc1*, *Il10*, and *Apoe* were considered to be M2 "signature" genes. We further observed a high heterogeneity of the relative expression of M2 genes within the clusters. Nevertheless, we were able to detect most of the mRNAs, which are categorized in Fig 2C and D as type 1 or 2, in the cluster 1 and 2 (Fig S4D). Examples for these mRNAs are *Retnla*, *Mamdc2*, *Kitl*, *Nes*, *Gypc*, *Proz*, *Chst11*, and *Mrc1*. M2 "signature" mRNAs and Type 1/2 mRNAs also showed the exact same pattern comparing IL4+IL13 and IL4+IL13+TNF stimulation (Fig S4E). In the "new" populations 0, 5, and 8, which appeared after TNF stimulation, a dominant gene expression signature of inflammatory-associated gene expression was evident, including increased TNF expression (Fig S4C). When considered with the bulk RNA sequencing, TNF had two main effects on IL4+IL13-stimulated M2 macrophages. First, the TNF-mediated inhibition of M2 gene expression is extraordinarily specific. Nearly all IL4+IL13-regulated gene expression is unaffected by TNF or cooperatively increased. Second, TNF elicits a population-level effect that regulates hundreds of genes, many of which are associated with inflammation. One of these targets is TNF itself, which may reflect feed-forward signaling to suppress selected M2 gene expression.

### TNF selectively controls M2 gene transcription

Based on the selective effects of TNF on IL4+IL13-induced mRNAs and that pSTAT6 activation was unaffected, we suspected TNF was

---

macrophages were analyzed after in vivo injection of solvent or Etanercept. Percentages of Arg1+ and RELMα+ cells were analyzed by flow cytometry analysis. See also Fig S1. All values are means ± SEM; *P < 0.05; ****P < 0.0001. Statistically significant differences were determined by one-way ANOVA with Tukey's correction or by paired t test. If not indicated otherwise superscripts show statistical significance compared with the control group. n = 3 biological replicates.

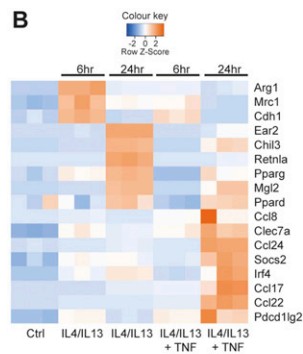

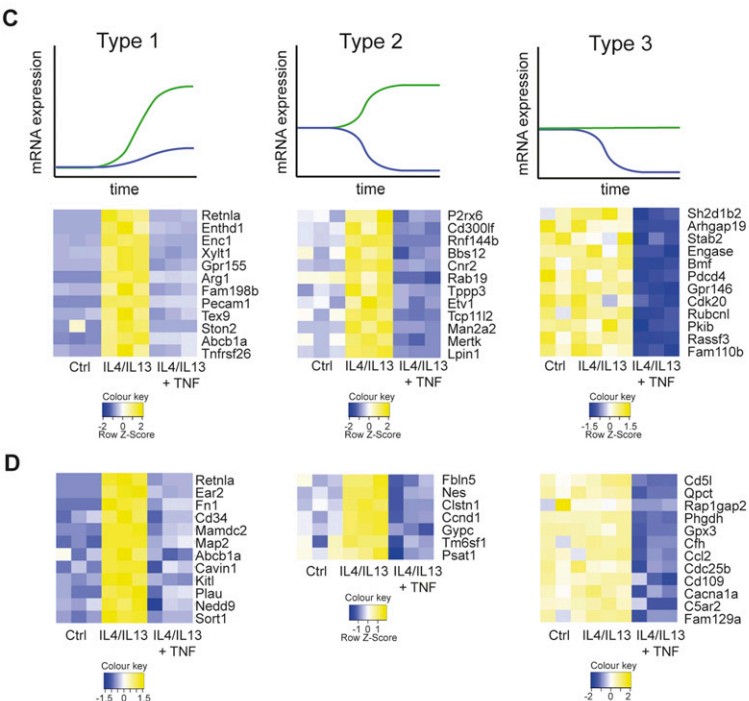

**Figure 2. TNF selectively blocks M2 gene expression.**
**(A)** RNA from wild-type BMDMs left untreated (Ctrl), stimulated with IL4/IL13, IL4/IL13 + TNF or TNF was used for RNAseq. Venn diagrams showing overlapping genes at 6 and 24 h. The diagram displays IL4/IL13 versus Ctrl up-regulated genes (green), TNF versus Ctrl up-regulated genes (white) and IL4/IL13 + TNF versus IL4/IL13 down-regulated genes (blue). See also Fig S2. **(B)** Heat map showing transcriptomic data from RNAseq analysis of Ctrl, IL4/IL13, and IL4/IL13 + TNF stimulated BMDMs over time. **(C, D)** Types of genes after 6 h (C) or 24 h (D) of IL4/IL13 (green line) or IL4/IL13 + TNF (blue line) stimulation. n = 3 biological replicates.

targeting M2 gene expression at the transcriptional level. To test this, we used primary transcript RT–PCR (PT RT–PCR) to estimate the rate of transcription of selected TNF-regulated genes by measuring the nascent unspliced mRNA in comparison with the conventional spliced transcript. The critical control for PT RT–PCR is the omission of reverse transcriptase in parallel matched samples to quantify contaminating genomic DNA (Murray, 2005). We added TNF 12 or 18 h after IL4+IL13 stimulation of BMDMs (time zero) and then analyzed all the samples at 24 h of IL4+IL13 stimulation. TNF inhibited the expression of the *Retnla* or *Fn1* PT (Figs 4A and B and S5A–D), indicating that a transcriptional process is likely the main mechanism TNF uses to repress M2 transcript accumulation. To gain insight into how M2 genes like Retnla are modified because of TNF treatment, we performed an ATACseq analysis to assess changes in chromatin accessibility. ATACseq analysis comparing IL4+IL13 or IL4+IL13+TNF treated wild-type BMDMs showed that IL4+IL13 stimulation caused increased chromatin accessibility compared with control macrophages. However, the addition of TNF substantially reversed this

chromatin accessibility (Fig 4C). M2 genes repressed by TNF (like *Clec10a* and *Mgl2*) had a decreased chromatin accessibility with IL4+IL13+TNF stimulation (Fig 4D). These findings suggest both IL4+IL13 and TNF have substantial and profound effects on the chromatin environment in macrophages.

### TNF signaling has a neutral effect on the metabolic state of M2 macrophages

A key question in the field of immunometabolism concerns the relationships between glycolysis, OXPHOS and gene and protein expression in different inflammatory states. Indeed, LPS stimulation, along with its subsequent autocrine-paracrine cytokine output including TNF, drives loss of mitochondrial function and enforcement of glycolysis, a state thought to be irreversible (Van den Bossche et al, 2017; van Teijlingen Bakker & Pearce, 2020). Within the transcriptome data of macrophages stimulated with TNF in the presence or absence of IL4+IL13, we noted many highly induced

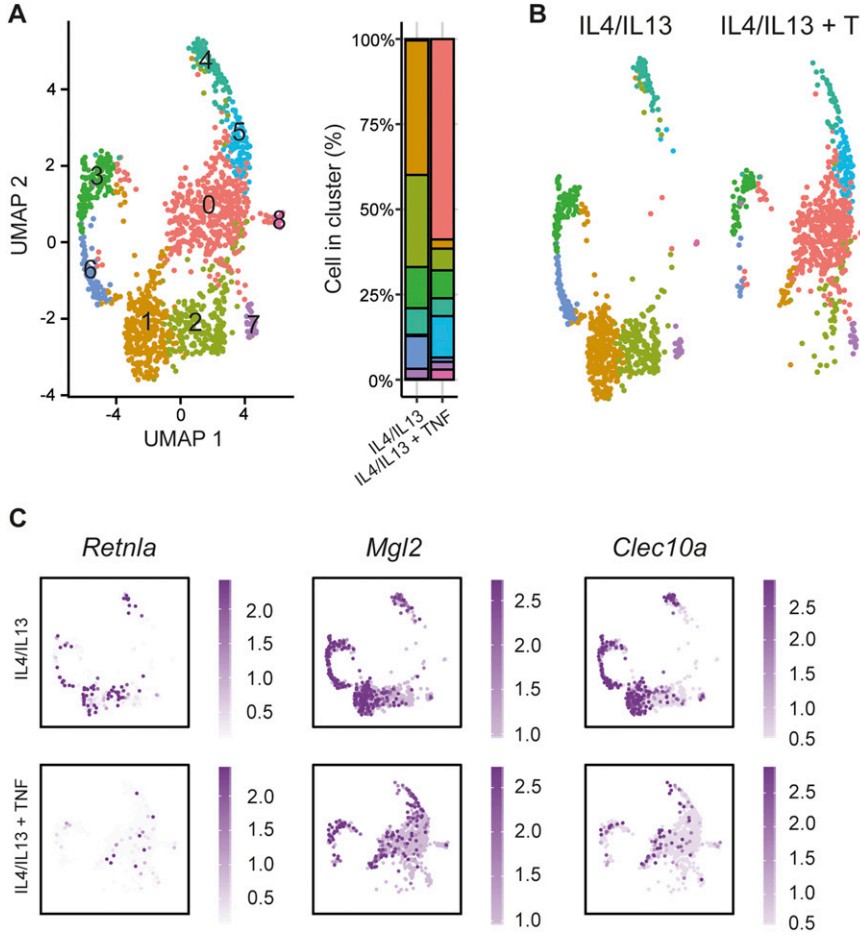

**Figure 3. Heterogeneity in TNF-responsive macrophages resolved at single cell resolution.**
**(A)** scRNA-seq analysis of wild-type BMDMs stimulated with IL4/IL13 or IL4/IL13 + TNF shown as a Uniform Manifold Approximation and Projection, highlighting identified clusters and cell distribution within clusters. **(B)** Uniform Manifold Approximation and Projection graph of A labelled according to the treatment. **(C)** Expression of M2 genes (*Retnla*, *Mgl2*, and *Clec10a*) in IL4/IL13 or IL4/IL13 + TNF treated samples.

transcripts encode metabolic modifiers associated with metabolic adaptation to inflammation, including Acod1 (also known as Irg1), Ass1 and Nos2, all enzymes which are associated with pro-inflammatory metabolism in macrophages (Fig S3C). These findings prompted us to investigate the effect of TNF on the metabolic status of M2 macrophages. We stimulated control or *Tnfr1a⁻/⁻* BMDMs with IL4+IL13, TNF or IL4+IL13+TNF and measured oxygen consumption rate (Fig S6A) and extracellular acidification rate (Fig S6B) and did not detected any differences. The addition of TNF, in combination with IL4+IL13 or alone, failed to stimulate nitrite ($NO_2^-$, which is a metabolite of NO) production in comparison to LPS treatment (Fig S6C), and the amount of Acod1, whereas detectable and TNFR1-dependent, was a fraction of that made following LPS stimulation (Fig S6D). Consistent with these findings, we found that pretreatment of control BMDMs with TNF for 24 h resulted in a comparable IL4+IL13-mediated induction of M2 genes, indicating the "neutrality" of TNF towards the bioenergetic status of IL4+IL13 treated macrophages (Fig S6E). Thus, TNF does not alter or inhibit the metabolic activity of M2 macrophages in contrast to the effects of LPS, which block OXPHOS, a process linked with inducible nitric oxide synthase/ nitric oxide (NO) and Acod1/itaconate (van Teijlingen Bakker & Pearce, 2020; Eming et al, 2021). Therefore, we concluded that TNF's selective effects on the M2 transcriptional program were unlikely

to be a consequence of modification of the essential function of OXPHOS in the M2 phenotype.

## Anti-M2 signaling can be mediated by TNF-independent inflammatory pathways

TNF signaling activates multiple pathways including death signaling via Ripk1, Ripk3, and caspase-8, NF-κB activation, and MAP kinase signaling. These pathways are common to other inflammatory receptors, including the Toll-like receptor and IL1 signaling pathways. We reasoned that we could indirectly discern information about how TNF controlled selective M2 gene expression by a comparative approach. When we stimulated BMDMs with IL1β or IL36, which signal through IL1R family receptors, *Retnla* expression was inhibited similarly to TNF (Fig 5A–C). Therefore, a pathway common to the IL1R and TNF receptors is activated to block M2 gene expression downstream of STAT6. Recent results showed the type 2 TNF receptor is involved in the control of M2 gene expression and indeed, loss of TNFR2 caused altered *Arg1* and *Mgl2* expression in M2 polarization conditions (Fu et al, 2021). However, when we used BMDMs lacking the TNFR1, TNF-mediated suppression of *Retnla* was completely lost (Fig 5D). Therefore, the anti-M2 effects of TNF in our

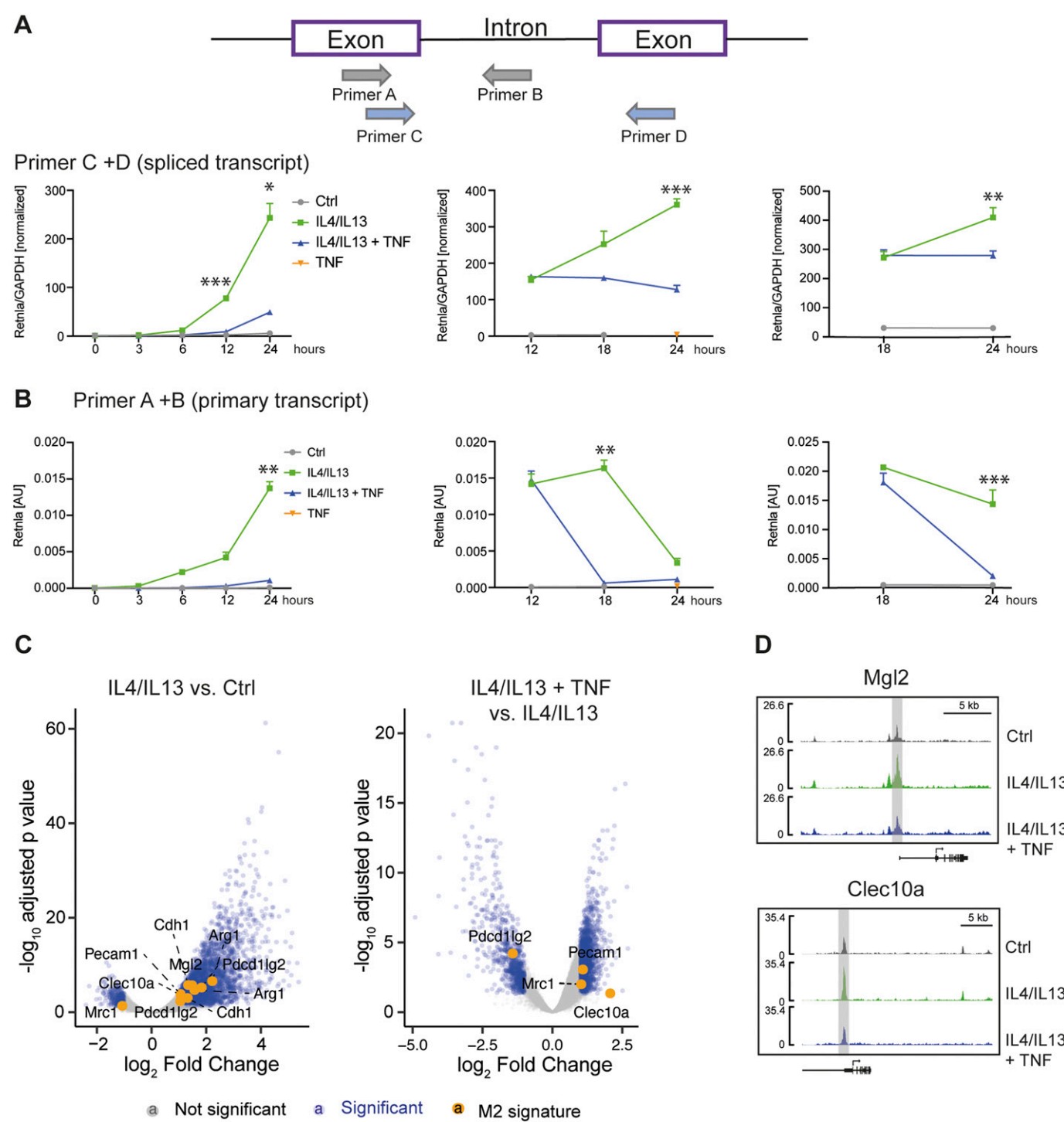

**Figure 4. TNF selectively controls M2 gene transcription.**
**(A)** Overview of the experimental design. RNA from wild-type BMDMs left untreated (Ctrl), stimulated with IL4/IL13, IL4/IL13 + TNF, or TNF was used for qRT-PCR. TNF was added at the same time like IL4/IL13 (left), 12 h after IL4/IL13 stimulation (middle) or 18 h afterwards (right). The last time point was always 24 h of IL4/IL13 stimulation. qRT-PCR for Retnla was analyzed. Data shown are the mean fold-increase of the Ctrl group. **(B)** Primary transcript qRT-PCR of the samples shown in (A). Superscripts show statistical significance between IL4/IL13 and IL4/IL13 + TNF. See also Fig S5. **(C)** Wild-type BMDMs were stimulated with IL4/IL13 or IL4/IL13 + TNF for 24 h or left untreated (Ctrl). Volcano plots of ATACseq data set showing IL4/IL13 versus Ctrl or IL4/IL13 + TNF versus IL4/IL13. Significantly changed sites are colored in blue and a selection of M2 genes are highlighted in orange. **(D)** Example of successfully identified accessible sites (ACS) in ATACseq data set. Significantly changed transcription sites of the ACS nearby Mgl$_2$ and Clec10a are highlighted in grey. n = 3 biological replicates.

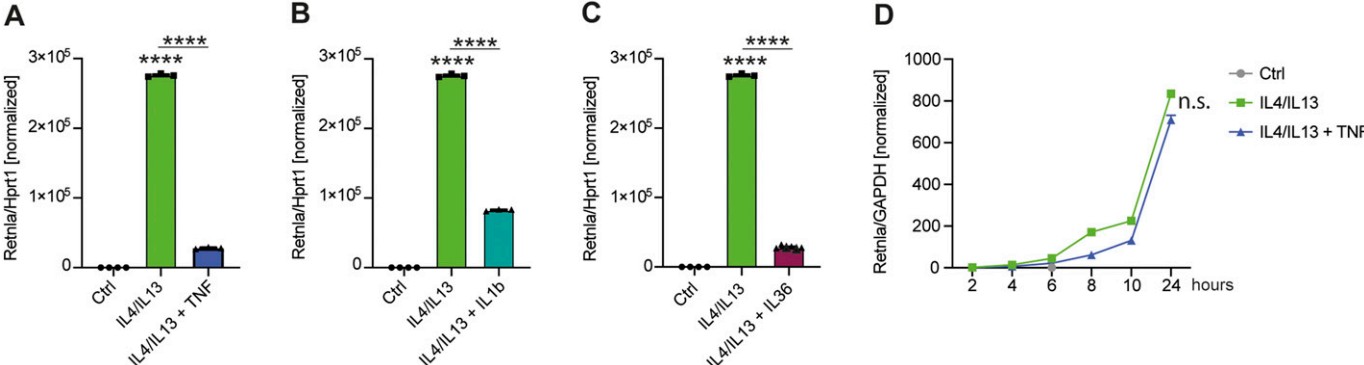

**Figure 5. Anti-M2 signaling can be mediated by TNF-independent inflammatory pathways.**
**(A, B, C)** RNA from wild-type BMDMs left untreated (Ctrl), stimulated with IL4/IL13, IL4/IL13 + TNF (A), IL4/IL13 + IL1b (B) or IL4/IL13 + IL36 (C) was used for qRT-PCR. Data shown are the mean fold-increase compared with the Ctrl group. **(D)** $TnfR1^{-/-}$ BMDMs were stimulated with IL4/IL13 or IL4/IL13 + TNF over time. RNA was isolated and analyzed for *Retnla* expression. Data shown are the mean fold-increase of the Ctrl group. All values are means ± SEM; ****$P$ < 0.0001. Statistically significant differences were determined by one-way ANOVA with Tukey correction (A, B, C) or by two-way ANOVA (D). If not indicated otherwise superscripts show statistical significance compared with the control group. n = 3 biological replicates.

system required the TNFR1 but not the TNFR2 and involved a signaling pathway common to the IL1R.

### TNF controls JNK signaling to a restricted set of Jun transcription factors

To evaluate transcription factors potentially regulated by TNF in the context of IL4+IL13 signaling, we performed a phospho-proteomic study. We detected 18,559 phosphopeptides of which 191 were significantly regulated comparing IL4+IL13 and IL4+IL13+TNF stimulated BMDMs across all time-points. Of these, 13 were downregulated (e.g., Csf1r or Afadin) with IL4+IL13+TNF compared with IL4+IL13 treated BMDMs. In general, the phosphoproteome changes were primarily linked to TNF stimulation and resulted in an upregulation of phosphopeptides with IL4+IL13+TNF (Fig 6A and Table S1). To link these observations of the effects of TNF on the M2 pathway, we first analyzed known TNF-dependent signaling pathways. Of the main signaling pathways activated by TNFR1, the anti-M2 effects of the Ripk1–Ripk3–Caspase-8 seemed unlikely to play a major role as (i) macrophages maintained their viability through all the stimulations and (ii) we found Ripk1 Ser25 phosphorylation was prominently detected in time-resolved phospho-proteomic analysis (Fig 6A). Ser25 phosphorylation inactivates the kinase activity of Ripk1 to suppress both necroptosis and apoptosis (Dondelinger et al, 2019). Furthermore, a prominent effect of NF-κB signaling also seemed unlikely as few canonical NF-κB targets (like Clec4e or Cxcl10) were activated by TNF in RNAseq experiments (with or without IL4+IL13 stimulation) (Fig S7A). The NF-κB targets were chosen according to the categorization of primary and secondary TLR-induced mRNAs, most of which are dependent to some degree on NF-κB signaling (Hargreaves et al, 2009; Ramirez-Carrozzi et al, 2009). Finally, the proteasome inhibitor MG132 failed to block the inhibitory effect of TNF on *Retnla* expression (Fig S7B) ruling out that the inhibitory effect of TNF is unlikely to be due to the proteosomal degradation of IκBα and subsequent activation of canonical NF-κB signaling. The analysis of the phosphoproteome data revealed early

TNF-induced phosphorylation events on four proteins involved in "transcription factor binding, transcription factor activity" (Gene Ontology Molecular Function) (Fig 6B). Two of these proteins are JunD and JunB. Consistently, previous datasets investigating the TNF phosphoproteome in various cell lines also revealed increased phosphorylation of these proteins (Tanzer et al, 2021). In early work on TNFR1 signaling, Karin and colleagues proposed that the MAP kinases of the Jun N-terminal kinase (JNK1 and JNK2) are activated in a sequence requiring TNFR1-associated death domain protein (TRADD), Fas-associated death domain protein (FADD), TRAF2, and upstream mitogen-activated protein kinase kinase kinase (MEKK) activation (Karin, 1995; Liu et al, 1996). JNK1/2 kinase activity culminates in the modulation of bZIP transcription factors, especially those of the Jun group (Liu et al, 1996). Based on the notion that JNK signaling was involved in the TNF-mediated control of M2 polarization and that multiple lines of evidence implicate JNK signaling to bZIP factors in macrophage activation, we performed experiments to investigate the relationships between TNF-regulated JNK signaling and the M2 pathway (Li et al, 1999; Jochum et al, 2001; Fontana et al, 2015; Hannemann et al, 2017; Fonseca et al, 2019).

In our ATACseq analysis, bZIP factor binding after TNF stimulation was significantly enriched over all other transcription factors (Fig 6C), and we observed that on average there was reduced accessibility of these predicted AP-1–binding sites (Fig 6D), suggesting increased occupancy by this recruited transcription factor after IL4+IL13+TNF compared with IL4+IL13 treated macrophages. AP-1 is a dimeric complex that is composed of members from the JUN (c-Jun, JunB, and JunD), FOS (c-Fos, FosB, Fra-1, and Fra-2) (Fig S7C), ATF or MAF protein families, which all show bZIP motifs (Shaulian & Karin, 2002; Eferl & Wagner, 2003). There are 45 bZIP transcription factors, which can form a variety of homo- and heterodimers (Newman & Keating, 2003). In macrophages, a restricted number of bZIP proteins including ATF3, Fos, FosL2, Jun, JunD, and JunB, coordinate the expression of thousands of genes in inflammation (Fonseca et al, 2019). However, when we screened our RNAseq data for the expression of the main bZIP family members, a restricted profile of

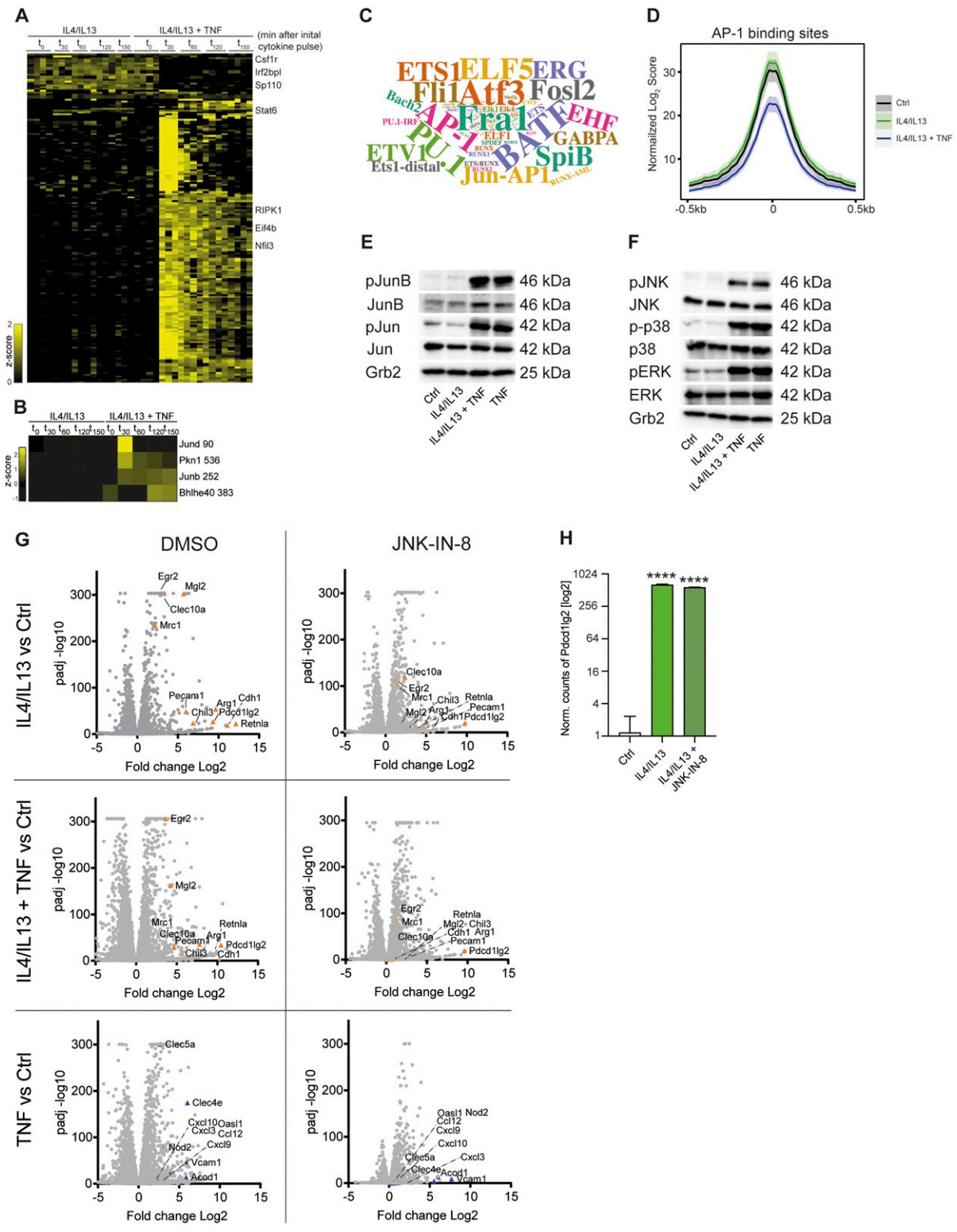

**Figure 6. JNK signaling has broad and profound effects on macrophage polarization.**
**(A)** Heat map of z-scored phosphosite intensities significantly changing in BMDMs treated with IL4/IL13 or IL4/IL13 + TNF for 0, 30, 60, 120, and 150 min (ANOVA, false discovery rate < 0.05). **(B)** Heat map of means of z-scored phosphosite intensities significantly changing in BMDMs treated as in (A) and filtered for GOMF "transcription factor binding, transcription factor activity." **(C)** Transcription factor enrichment analysis shown as a word cloud where size of name is proportional to the number of gene sets in transcriptional network clusters associated with specific transcription factor. **(D)** HINT-ATAC analysis of predicted AP-1 binding sites. **(E, F)** Wild-type BMDMs were stimulated with IL4/IL13, IL4/IL13 + TNF or TNF for 15 min and whole-cell lysates were isolated and analyzed by Western blotting for the indicated proteins. **(G)** Volcano plots of transcriptomics data set from profile GSE185380. The stimulated groups were compared to the responding Ctrl group. **(H)** *Pdcd1lg2* gene analysis of this data set.

bZIP factors including JunB was expressed and regulated by TNF in combination with IL4+IL13 signaling at the mRNA level (Fig S7D). We also observed TNF-dependent phosphorylation of JunB and c-Jun (Fig 6E). In these experiments, antibodies to JunB Thr102/Thr104 or c-Jun Ser63 were used, which are the main phospho-sites targeted by JNK1/2 (Li et al, 1999; Davis, 2000). The combination of IL4+IL13+IL1β, which already showed comparable results to the IL4+IL13+TNF stimulated cells in the inhibition of *Retnla* expression (Fig 5B), was also able to increase the phosphorylation of JunB in a TNFR1-dependent way (Fig S7E). Consistent with the activation of the bZIP factors, the main MAP kinases (extracellular signal-regulated kinase [ERK], p38, and JNK) were activated by TNF in the presence or absence of IL4+IL13 (Fig 6F). The downstream targets of ERK, c-Fos, and FosB, were mainly regulated by TNF in comparison with JunD, which were up-regulated by the combination of IL4+IL13+TNF (Fig S7F and G).

As JNK, JunB, and c-Jun were circumstantially linked to the TNF anti-M2 pathway, we next considered different ways to perturb these factors. Previous genetic studies established that a JNK1/2 (encoded by Mapk8 and Mapk9) double knockout is lethal (Kuan et al, 1999; Sabapathy et al, 1999). Furthermore, both *Jun* and *Junb* deficient mice die in embryogenesis (Hilberg et al, 1993; Johnson et al, 1993; Schorpp-Kistner et al, 1999). We therefore attempted to delete *Junb* by lentivirus-mediated Cripsr-Cas9 genome modification using Cas9+ bone marrow progenitors. No macrophages were recovered from these experiments, indicating an essential requirement for JunB in macrophage development. We next turned to chemical perturbation, taking advantage of a family of JNK inhibitors that irreversibly inactivate JNK kinase activity by forming a covalent bond to Cys154 in the active site of both enzymes (Zhang et al, 2012). When we added JNK-IN-8 to BMDM cultures at the beginning of CSF1-mediated proliferation and differentiation, no macrophages were recovered. This was an identical observation to Himes et al (2006) who used different modes of JNK inhibition and concluded that JNK signaling was required for CSF1R activity and thus the normal development of BMDMs (Himes et al, 2006). To bypass the effect of JNK inhibition on CSF1-mediated macrophage development, we found that when we added JNK-IN-8 after macrophages had developed and proliferated for ~7 d, TNF-regulated p-c-Jun and p-JunB were completely inhibited, providing us with a means to test the role of JNK signaling in the TNF anti-M2 pathway (Fig S7H).

### JNK signaling has broad and profound effects on macrophage polarization

We next tested if pharmacological JNK inhibition after BMDM development was completed would block the anti-M2 activity of TNF and "mirror" the phenotype of the TNFR1 knockout. Such a hypothesis would be supported in the cases where JNK was an obligatory factor in the TNFR signaling pathway but not required for IL4R signaling. We compared the effects of JNK-IN-8 versus a noncovalent ATP competitive JNK inhibitor (SP600125) on JunB and

c-Jun phosphorylation after BMDM stimulation with IL4+IL13 or IL4+IL13 with concurrent TNF stimulation. The idea behind this was to use SP600125 to transiently inhibit JNK and evade the profound effects of JNK-IN-8 on CSF1R signaling. However, SP600125 had minimal effects on JunB and c-Jun phosphorylation compared with JNK-IN-8, which by contrast was a complete inhibitor of the downstream targets of JNK (Fig S7H).

Next, we used JNK-IN-8 to survey the extent of the JNK-dependent transcriptional changes after BMDMs stimulation with IL4+IL13, TNF, or IL4+IL13+TNF. As expected, the DMSO controls resulted in a similar pattern of TNF and IL-4+IL13 effects as described above and served as internal controls for the fidelity of the analysis. JNK-IN-8, however, caused a striking and broad effect on activated macrophage gene expression. First, nearly all TNF-regulated transcription was ablated, consistent with the proposed primacy of JNK signaling from the TNFR1 (e.g., Type 2 genes such as *Mertk* or Type 3 genes such as *Arhgap19*) (Liu et al, 1996). Second, JNK-IN-8 had an unexpected effect on part of IL4+IL13-induced transcription; almost all M2 mRNAs negatively regulated by TNF were also inhibited by JNK-IN-8 (Type 1 genes, e.g., *Retnla*) (Fig 6G). These findings are in agreement with previous observations that MKK7, which activates JNK, is enriched in phagosomes of IL4-activated macrophages (Hao et al, 2017; Guo et al, 2019). Thus, even though IL4+IL13 stimulation showed minimal activation of the JNK pathway by measuring pathway phosphorylation (compared to TNF) (Fig 6E and F), JNK signaling to downstream transcription factors was obligatory to a fraction of the M2 program and thereby implicating JNK in the expression of all three observed TNF-regulated gene types (Fig 2C and D). Third, some mRNAs strongly induced by IL4+IL13 were unaffected by JNK inhibition; for example, the expression of the PD-L2 mRNA (Fig 6H). JNK signaling therefore controls overlapping modes of both macrophage development and polarization by affecting different signaling pathways at different points.

## Discussion

Our experiments were initiated with the idea of establishing the mechanistic basis of the anti-M2 effects of TNF; in part, we reasoned that as anti-TNF biologics have a vast spectrum of activities in RA and IBD patients, we should be able to elucidate a signaling framework that connects successful anti-TNF therapy with the number and function of M2 macrophages. In previous experimental systems, we and others showed that many hallmark M2 genes were negatively regulated by TNF (Kroner et al, 2014; Kratochvill et al, 2015b; Schleicher et al, 2016; Li et al, 2017, 2021). Here, we used more general approaches to evaluate the global effects of TNF on M2 gene expression. We found that the effects of TNF on M2 pathway gene expression are highly gene specific. Although IL4+IL13 activates >800 mRNAs by 24 h, TNF negatively regulates ~20 of these mRNAs. TNF acts on this restricted cohort of genes at the

---

All values are means ± SEM; ****$P < 0.0001$. Statistically significant differences were determined by one-way ANOVA with Tukey correction. Superscripts show statistical significance compared with the control group. n = 3 biological replicates.

transcriptional level with differential effects; expression of some IL4+IL13 targets such as *Retnla* were completely suppressed by TNF, whereas other targets were partly inhibited (*Mgl2*) or blocked in a time-dependent way (*Arg1*). TNF is therefore a context-specific modulator of M2 macrophages rather than a pan-M2 inhibitor.

So far, accurate measurements of inflammatory gene expression in resolving information have been made in the skin wound healing model (Knipper et al, 2015). This model follows a predictable and reproducible time dependency from the initial inflammatory reaction, which shifts to wound healing and finally scar formation. TNF, along with other inflammatory mediators, peaks in the first days after wounding and precedes the emergence of macrophage gene expression linked to M2 macrophages (Willenborg et al, 2012; Knipper et al, 2015). In this context, we found that Etanercept augments *Retnla* expression. However, in pathology where too much TNF is driving the pathology, anti-TNF has beneficial effects; most likely through induction of M2, as demonstrated in this article. Thus, we can place our findings about the specificity of the anti-TNF M2 pathway in clinical context; the outcomes of anti-TNF biologic therapy in RA and IBD in terms of enhancement of the number and function of M2 macrophages are likely variable and patient specific. An additional finding in this regard concerns the fact that cytokines such as IL1$\beta$ and IL36 can achieve similar effects to TNF. Although we used this information to understand the signaling pathway that controlled M2 expression, a broader interpretation is that inflammatory lesions replete in combinations of IL1$\beta$, IL36, and TNF may have additive anti-M2 effects.

At the cellular level, our scRNAseq data highlighted that IL4+IL13 stimulated cultures of BMDMs are heterogenous and that there is no clear pattern of M2 gene expression within the cluster distribution, nevertheless we observed the IL4+IL13 stimulated group mainly in the cluster 1, 2, and 6. Our experiments agree with recent work that shows that a seemingly homogenous bone marrow culture expanded in CSF1 or GMCSF, which was unstimulated, has an underlying heterogeneity in gene expression and cytokine responsiveness (Helft et al, 2015, 2016; Muñoz-Rojas et al, 2021). We observed that although all cells in our scRNAseq analysis expressed type 1 and 2 TNFR, some cells were unresponsive to the effects of TNF on M2 gene expression. These results suggest that some degree of "pre-programming" for differential responsiveness to cytokines occurs in bone marrow progenitors, which is then reflected in the heterogenous responses detected at the single cell level. Such heterogeneity in bone marrow-derived macrophage progenitors could be a contributing factor to the wide variations of the therapy efficacy with anti-TNF biologics. We further found that TNF is not sufficient enough to alter macrophage metabolism and thereby intersects with M2 macrophage gene expression compared to LPS stimulation. A systematic comparison of how different pro-inflammatory macrophage stimulation modalities act would likely shed light on the molecular pathways that alter macrophage metabolism.

Interrogation of the JNK and Jun family of bZIP proteins in macrophages has intrinsic challenges. Mice lacking JNK1 and JNK2, c-Jun, and JunB are all embryonic lethal and JNK inhibition during macrophage development is similarly lethal. Conditional deletion of genes encoding JunB and JNK1/2 has revealed different effects on macrophage polarization pathways (Han et al, 2013). However, the Cre deleter used in these cases was LysM-Cre, which produces highly locus-specific and often incomplete deletion (Kratochvill et

al, 2015a; Murray, 2017). Given that our results with JNK-IN-8 inactivation of JNK1/2 produced substantial and broad effects on both TNF and IL4+IL13 gene expression and completely inhibited macrophage development from the CSF1 pathway, most likely the phenotypes described in the JunB and JNK1/2 conditional mice reflect a "knockdown" of each factor rather than a complete deletion. When considered with the fact that Jun family bZIP proteins regulate thousands of macrophage genes in complex, overlapping and partly redundant ways during both development and active inflammation (Fonseca et al, 2019), the perturbation of this pathway may seem intractable from a genetic perspective. However, a different way to approach this problem is through the use of targeted protein degradation (after macrophage development) to take advantage of the specific phosphorylation events that occur downstream of each signaling pathway. For example, a PROTAC drug that recognized phosphorylated JunB and directed it to proteosomal degradation at specific points after TNF stimulation would in theory enable time-resolved elimination of active JunB with limited effects on cell viability. In our view, this is the most feasible way to perturb a dynamically and spatially modulated pathway required for multiple signaling events and development.

## Materials and Methods

### Animals

For BMDM culture, male C57BL/6J (Wt), *Stat6*$^{-/-}$ (002828; purchased from The Jackson Laboratory), and *Tnfr1a*$^{-/-}$ mice were mated and maintained under specific pathogen-free conditions at the animal facility of the Max Planck Institute of Biochemistry. Both *Stat6*$^{-/-}$ and *Tnfr1a*$^{-/-}$ mice were on a C57BL/6J background. Use of mice for breeding and organ isolation was approved by the Government of Upper Bavaria. For the in vivo wound healing model 10–12 wk old female wild-type BALB/c mice were used and were approved by the North Rhine–Westphalian State Agency for Nature, Environment, and Consumer Protection (LANUV) and the University of Cologne (reference number 84-02.04.2017.A019). Mice, which were used for the isolation of peritoneal macrophages, were maintained in specific pathogen-free conditions under protocols approved by the animal care committees of the Regierungspräsidium Freiburg and the Institutional Animal Care and Use Committee of Johns Hopkins University, in compliance with all relevant ethical regulations.

### Primary cell culture

BMDMs were grown in DMEM (41966-029; Thermo Fisher Scientific) composed of 10% FBS (S0115; Biochrom), penicillin/streptomycin and CSF1. The BMDMs were cultured for 7 d on 150 mm TC-treated culture dishes (430599; Corning) in the presence of endotoxin-free baculovirus-expressed recombinant human CSF1 generated in-house. Every other day 3 ml of new CSF1-containing medium was added to the culture. On day 7, BMDMs were harvested with a cell scraper (422-83.1830; Sarstedt). After counting, the cells were seeded in the presence of 100 ng/ml CSF1 and stimulated the next day. All cell culture procedures were carried out at 37°C with 5% $CO_2$.

## Stimulation of BMDMs

Where indicated, BMDMs were stimulated with 10 ng/ml IL4 (produced in insect cells), 10 ng/ml IL13 (210-13; Peprotech), 10 ng/ml TNF (315-01A; Peprotech), 10 ng/ml IL1$\beta$ (401-ML/CF; R&D System), 33.3 ng/ml IL36$\alpha$ (7059-ML/CF; R&D Systems), 33.3 ng/ml IL36$\beta$ (7060-ML/CF; R&D Systems), 33.3 ng/ml IL36$\gamma$ (6996-IL/CF; R&D Systems), 5 ng/ml LPS from *Escherichia coli* O111:B4 (L4391; Sigma-Aldrich), 10 μg/ml Etanercept (Erelzi; Sandoz), 20 μM JNK-IN-8 (SML1246; Sigma-Aldrich), and 10 μM SP600126 (BML-EI305; Enzo). TNF was always added at the same point as IL4+IL13 (except Fig 4A and B).

## RNA isolation and analysis

BMDMs were lysed in TRIzol (15596018; Invitrogen) for subsequent RNA isolation according to the manufacturer's instructions. cDNA was synthesized using SuperScript IV reverse transcriptase (18090050; Invitrogen), random hexamers (N8080127; Invitrogen) and oligo(dT) primers (N8080128; Invitrogen), and analyzed by qRT-PCR. Retnla primers were purchased from QIAGEN. All values were normalized to GAPDH (4352932E; Applied Biosystems). Primer sequences can be found in Table 1.

## Bulk RNAseq

RNA samples were subjected to mRNA-seq and quantified using Qubit 2.0 according to the manufacturer's instructions. RNA was polyA enriched. Libraries were prepared and sequenced on an Illumina NextSeq 500 platform. Raw mapped reads were processed in R to determine differentially expressed genes and generate normalized read counts. RNAseq data are available in GSE169348 and GSE185380.

## Flow cytometry

BMDMs were suspended in FACS buffer (PBS + 1% FBS) and incubated with Fc block (anti-mouse CD16/32; BioLegend). Cell surface marker were stained with anti-F4/80 (123117; BioLegend) and for live/dead staining with LDFA (L34957; Thermo Fisher Scientific). Afterwards, cells were washed, permeabilizated with Foxp3/Transcription Factor Staining Buffer Set (00-5523-00; Thermo Fisher Scientific), blocked with 2% normal mouse serum (Invitrogen) and stained for intracellular Arg1 (17-3697-82; Thermo Fisher Scientific) and Retnla (500-P214-50; PeproTech and P10994; Thermo Fisher Scientific) expression. Cells were washed twice and then resuspended for analysis on a BD LSR Fortessa FACS flow cytometer. Gating strategy is shown in Fig S1A. Data analysis was carried out using FlowJo software.

## In vivo models

### SPMs

Healthy wild-type mice were injected i.p. with PBS or 12 mg/kg bodyweight Etanercept three times every other day for a total of 6 d. On day 6, the mice were euthanized and the peritoneum was flushed with PBS. SPMs were analyzed by flow cytometry using the gating strategy shown in Fig S1G.

### Wound healing model

**Excisional punch injury** 10–12-wk-old female wild-type BALB/c mice were anesthetized by i.p. injection of Ketanest/Rompun (Ketanest S; Park Davis; Rompun 2%; Bayer). Littermates of the same sex were randomly assigned to experimental groups. The back skin was shaved, and full-thickness punch biopsies were created on the back using a standard biopsy puncher (Stiefel). Wounds were excised at indicated time points after injury and processed to generate single-cell suspensions for flow cytometry following an established protocol (Willenborg et al, 2012; Knipper et al, 2015).

**Administration of Etanercept** Mice were treated i.p. with 12 mg/kg bodyweight Etanercept (Erelzi; Sandoz) or vehicle (NaCl) in a total volume of 200 μl every third day starting 3 d prior wounding and were continued every 3 d until the mice were euthanized.

**Flow cytometry** Single-cell suspensions of wound tissue were prepared by a combination of enzymatic digestion (Liberase Blendzyme; Roche Applied Science) and mechanical disruption (Medimachine System; BD Biosciences). Excised wound tissue was sectioned with a scalpel, placed in DMEM medium with 30 μg/ml Liberase TM Research Grade (5401119001; Roche Applied Science), and incubated at 37°C for 90 min (shaking). Digested wound tissue was mechanically disrupted for 5 min using the Medimachine System. Cells were passed through 70 and 40 μm cell strainer, and washed with PBS/1% BSA/2 mM EDTA. Fc receptors were blocked with mouse SeroBlock FcR (CD16/CD32; 16-0161-82, 1:50; eBioscience) and cells were stained with fluorescein isothiocyanate-conjugated anti-CD45 (11-0451-82, 1:200; eBioscience), allophycocyanin conjugated anti-CD11b (17-0112-82, 1:400; eBioscience), and phycoerythrin-conjugated anti-F4/80 (MCA497PE, 1:50; Bio-Rad Laboratories) in PBS/1% BSA/2 mM EDTA. Dead cells were excluded using 7 amino-actinomycin D (00-6993-50, 1:40; eBiosciences). Cells were sorted using a FACSAria III cell sorting system (BD) equipped with FACSDiva software (BD). Sorting strategy is shown in Fig S1F.

**Table 1. Primer sequences of qRT-PCR primer.**

| Name | Sequence |
| --- | --- |
| Abcb1a fw | GCGGAGTCAGACAGAACAAGA |
| Abcb1a rv | ATTCCCCCTTTTATCTGAATGCTT |
| Arg1 fw | CATGGGCAACCTGTGTCCTT |
| Arg1 rv | CGCAAGCCAATGTACACGAT |
| Fbln5 fw | TGCAGACAGAGACGCATGATA |
| Fbln5 rv | CCAGGTCAAAGCCGTTTGTG |
| Pdcd4 fw | CTGTGCCCACCAGTCCAAAA |
| Pdcd4 rv | GCCTGCACCACCTTTCTTTG |
| Qpct fw | GGACGCAGGAGAAGCACATC |
| Qpct rv | GGGAGTCCTACTCAGGAAGGT |
| Rnf144b fw | TTCAGCTCCAGCCAGGGATAG |
| Rnf144b rv | GGGTTTTCCGCAGTCATGGT |

**Quantitative real-time PCR analysis** For quantitative real-time PCR analysis, $2 \times 10^5$ flow cytometry-sorted CD45$^+$CD11b$^+$F4/80$^+$ wound macrophages were lysed in RLT lysis buffer (79216; QIAGEN) and total RNA was isolated using the RNeasy Micro Kit (QIAGEN) according to the manufacturer's instructions. To obtain one data point at 2, 4, and 7 d post injury (dpi), four wounds from one mouse were pooled; at 14 dpi eight wounds from two mice were pooled. Reverse transcription of isolated RNA was performed using the High Capacity cDNA RT Kit (4368814; Thermo Fisher Scientific). Amplification reactions (triplicates) were set up using the PowerSYBR Green PCR Master Mix (4368577; Applied Biosystems) and quantitative real-time PCR (qRT-PCR) was validated with the QuantStudio 5 real-time PCR system (Thermo Fisher Scientific). The comparative method of relative quantification ($2^{-\Delta\Delta Ct}$) was used to calculate the expression level of the target gene normalized to *Rps29*. Primer sequences can be found in Table 2.

### SmartSeq scRNAseq

Wild-type BMDMs were stimulated with 10 ng/ml IL4, IL13, and TNF. Cells were scraped and stained for live/dead staining with LDFA. The scRNAseq workflow was based on the previously described Smart-seq2 protocol (Picelli et al, 2013) with the following modifications. Single cells were flow-sorted into 384-well plates containing 0.5 μl of nuclease-free water with 0.2% Triton X-100 and 1 U murine RNase inhibitor (M0314S; New England Biolabs), centrifuged, and frozen at –80°C. After thawing, 0.5 μl of the primer mix (5 mM dNTP [Invitrogen], 0.25 μM oligo-dT primer (C6-aminolinker-AAGCAGTGGTATCAACGCAGAGTCGACTTTTTTTTTTTTTTTTTTTTTTTTTT TTTTTTVN, where N represents a random base and V any base beside thymidine, and 1 U murine RNase inhibitor) was added to each well. The reverse transcription reaction was performed with 1.6 μM of following template-switching oligonucleotides (AAGCAGTGGTAT-CAACGCAGAGTACATrGrGrG, where rG stands for ribo-guanosine as described [Picelli et al, 2013]), but with final concentrations of RNase inhibitor and Superscript II of 2.5 and 23 U, respectively, at 42°C for 90 min, followed by an inactivation step at 70°C for 15 min. For amplification, Kapa HiFi HotStart ReadyMix (7958927001; Roche) at a final 1× concentration and 0.1 μM UP-primer (AAGCAGTGGTATCAACGCA-GAGT) under following cycling conditions: initial denaturation at 98°C for 3 min, 22 cycles (98°C 20 s, 67°C 15 s, 72°C 6 min) and final elongation at 72°C for 5 min. The number of pre-amplification PCR cycles was increased to 22 to ensure there was sufficient cDNA for downstream analysis. The amplified cDNA was purified using Sera-Mag SpeedBeads (GE17152104010150; GE Healthcare) resuspended in a buffer consisting of 10 mM Tris, 20 mM EDTA, 18.5% (wt/vol) polyethylenglycol (PEG) 8000 and 2 M sodium chloride solution. DNA was eluted in 12 μl nuclease-free water. The concentration of samples was measured using a plate reader (Infinite 200 PRO; Tecan) in 384 well black, flat-bottom, low-volume plates (Corning) using an AccuBlue Broad

Range kit (31007; Biotium). Then, 0.7 ng of pre-amplified cDNA was used for library preparation (TruePrep DNA library preparation kit V2 for Illumina, TD202; Vazyme) in a 1-μl reaction volume. Illumina indices were added during the PCR reaction (72°C for 3 min, 98°C for 30 s, 12 cycles of [98°C for 10 s, 63°C for 20 s, and 72°C for 1 min], and 72°C 5 min) with 1 × KAPA Hifi HotStart ReadyMix and 0.33 μM of dual indexing primers. After PCR, the libraries were quantified with AccuBlue Broad Range kit, pooled in equimolar amounts, and purified twice with Sera-Mag SpeedBeads. The libraries were sequenced on the Novaseq 6000 Illumina platform to obtain 50 bp paired-end reads aiming at an average sequencing depth of 0.5 million reads per cell.

### ScRNAseq analysis

Fragments were aligned to the mouse reference genome mm10 using the aligner gsnap (2019-06-10) with Ensembl 98 splice sites as support (Wu & Watanabe, 2005; Wu & Nacu, 2010). Uniquely mapped fragments were compared based on their overlap to Ensembl 98 gene annotations using featureCounts (v1.6.3) (Liao et al, 2014) to create a table of counts per gene and cell. Read count matrices were processed, analyzed and visualized in R v. 4.0.0 (R Core Team, 2013; http://www.R-project.org) using Seurat v. 3 (Stuart et al, 2019) with default parameters in all functions, unless specified. Poor quality cells, with low total unique molecular identifier counts and high percent mitochondrial gene expression or reads mapping to external RNA control consortium spike-in controls, were excluded. Filtered samples were normalized using a regularized negative binomial regression (SCTransform) (Hafemeister & Satija, 2019) and integrated with the reciprocal principal component analysis approach followed by mutual nearest neighbors, using 50 principal components. Integrated gene expression matrices were visualized with a Uniform Manifold Approximation and Projection (McInnes et al, 2018) as a dimensionality reduction approach. Resolution for cell clustering was determined by evaluating hierarchical clustering trees at a range of resolutions (0–1.2) with Clustree (Zappia & Oshlack, 2018), selecting a value inducing minimal cluster instability. Differentially expressed genes between clusters were identified as those expressed in at least 25% of cells with a greater than +0.25 log fold change and an adjusted *P*-value of less than 0.01, using the FindMarkers function in Seurat v.4 with all other parameters set to default. Ribosomal protein genes were excluded from results. Gene set scores were calculated using the AddModuleScore function in Seurat v.4 with default parameters. Briefly, the average expression levels of each identified gene set were calculated on a single cell level and subtracted by the aggregated expression of randomly selected control gene sets. For this purpose, target genes are binned based on averaged expression, and corresponding control genes are randomly selected from each bin.

### PT qRT-PCR

PT qRT-PCR was done like previously described (Murray, 2005). Wild-type BMDMs were lysed by using Trizol. Lysates were treated with RNase-free DNase (M6101; Promega) for 30 min using 1 U of enzyme. Afterwards, RNA was phenol/chloroform-extracted and resuspended to a final concentration of 1.5 μg. Reverse transcription was performed by using SuperScript IV according to the manufacturer's instructions. Identical

**Table 2. Primer sequence used for qRT-PCR analysis of wound healing model.**

| Gene | Forward | Reverse |
|------|---------|---------|
| *Rps29* | GGTCACCAGCAGCTCTACTG | GTCCAACTTAATGAAGCCTATGTCC |
| *Retnla* | TATGAACAGATGGGCCTCCT | GGCAGTTGCAAGTATCTCCAC |

**Table 3. PT qRT-PCR primer sequences.**

| Name | Sequence |
|------|----------|
| Retnla_A | AGCTGATGGTCCCAGTGAAT |
| Retnla_B | GTCAAGAAGGCAGGGATGAA |
| Retnla_C | CCCTTCTCATCTGCATCTCC |
| Retnla_D | AGGAGGCCCATCTGTTCATA |
| Fn1_A | ACGAGGAGGGACATATGCTG |
| Fn1_B | ATGAGGCAGGTTTGGAGAGA |
| Fn1_C | TCTGCAGAGGTTGACAGTGC |

samples from each time point were processed in the absence of reverse transcriptase and served as controls for genomic DNA contamination. cDNA was subjected to qRT-PCRs (Primer C + D) or using primer pairs designed for PT qRT-PCR (Primer A + B). Relative quantities of the samples were determined against a diluted standard of genomic DNA. All primers sequences are listed in Table 3.

### Immunoblotting

Lysates were prepared from BMDMs on ice in RIPA buffer (ab156034; Abcam) containing protease and phosphatase inhibitors (78444; Thermo Fisher Scientific). Proteins were separated on Tris–HCl gradient gels (5678085 or 5678084; Bio-Rad Laboratories) and transferred to nitrocellulose (N010600001; Amersham). Membranes were blocked in 5% BSA (A2153; Sigma-Aldrich) and probed with primary antibodies overnight at 4°C. Anti-pStat6 (56554), Stat6 (5397), pJNK (9251), JNK (9252), pERK (9101), ERK (9102), p-p38 (9211), p38 (9212), Nos2 (2977), pJunB (8053), JunB (3753), p-cJun (9261), and cJun (9165) antibodies were obtained from Cell Signaling Technology. Irg1 antibody (ab222411) was purchased from Abcam. As loading control anti-Grb2 antibody (610112) was used from BD. Membranes were washed and probed with secondary antibodies at a 1:10,000 dilution and developed using chemiluminescence reagents (34580; Thermo Fisher Scientific).

### Seahorse bioenergetic measurements

Real-time oxgen consumption rate and extracellular acidification rates measurements were made with a Seahorse XF HS Mini Analyzer (Agilent Technologies). $0.04 \times 10^6$ BMDMs were plated into each well of Seahorse cell culture plates and preincubated at 37°C for a minimum of 45 min in the absence of $CO_2$ in Seahorse XF DMEM medium (103575-100; Agilent Technologies) supplemented with glutamine, glucose and pyruvate. After analysis with the Seahorse Cell Mito Stress Test Kit (103010-100; Agilent Technologies), normalization was made by protein quantification.

### Nitric oxide measurement

Nitric oxide was measured with Griess reagent according to Stuehr and Nathan (1989).

### ATACseq

Libraries were prepared using the Nextera DNA library Prep Kit (Illumina) adapting a published protocol (Buenrostro et al, 2015). Briefly, $5 \times 10^4$ BMDMs treated for 24 h were washed in PBS and then lysed in 10 mM Tris–HCl, pH 7.4, 10 mM NaCl, 3 mM $MgCl_2$, and 0.1% Igepal CA-630 (all Sigma-Aldrich). Nuclei were spun down and then resuspend in 25 $\mu$l TD (2× reaction buffer), 2.5 $\mu$l TDE1 (Nextera Tn5 Transposase) and 22.5 $\mu$l nuclease-free water, and incubated for 30 min at 37°C. DNA was purified with the QIAGEN MinElute PCR Purification Kit (28004; Thermo Fisher Scientific). PCR amplification was performed with the NEBNext High-Fidelity 2× PCR Master Mix (M0541L; New England Biolabs) using custom Nextera PCR Primers containing barcodes. Adaptors were removed with AMPure XP beads according to manufacturer's protocol. Libraries were quantified with the Qubit and submitted for sequencing with a HISeq 3000 (Illumina) by the staff at the Deep-sequencing Facility at the Max-Planck-Institute for Immunobiology and Epigenetics. Sequenced samples were trimmed with Trimmomatic (Bolger et al, 2014) and mapped using Bowtie2 (Langmead & Salzberg, 2012). Coverage files were generated with deepTools (Ramírez et al, 2016). Open chromatin was detected with MACS2 (Zhang et al, 2008), whereas differences between treatments were determined using DiffBind (Ross-Innes et al, 2012) with at least twofold change in accessibility and a false discovery rate (FDR) lower than 0.05. For visualization only, replicate mapped files were merged with Sequence Alignment/Map tools and coverage files were generated with deepTools and visualized alongside coverage files on Integrative Genomics Viewer (Robinson et al, 2011). Bed files were analyzed with Bedtools. Transcription factor enrichment was visualized with wordcloud2 (https://cran.r-project.org/web/packages/wordcloud2/vignettes/wordcloud.html), with word size proportional to enrichment $-\log_{10}$ P-value. Coverage tracks were visualized with Spark (Kurtenbach & Harbour, 2019 Preprint). Comparison of accessibility of specific transcription factor sites across treatments was performed with deepStats (Gautier RICHARD. [2019, August 6]. gtrichard/deepStats: deepStats 0.3.1 [Version 0.3.1]. Zenodo. https://doi.org/10.5281/zenodo.3336593).

### Phosphoproteomics

#### Phosphoenrichment protocol

To enrich for phosphorylated peptides, we applied the Easy Phos protocol developed in the Mann laboratory (Humphrey et al, 2015, 2018). In short, BMDMs were stimulated, washed three times with ice-cold TBS, lysed in 2% sodium deoxycholate and 100 mM Tris–HCl (pH 8.5), and boiled immediately. After sonication, protein amounts were adjusted to 1 mg using the BCA protein assay kit. Samples were reduced with 10 mM tris(2-carboxy(ethyl)phosphine), alkylated with 40 mM 2-chloroacetamide and digested with trypsin and lysC (1:100, enzyme/protein, w/w; WAKO Chemicals) overnight. Isopropanol (final conc. 50%), TFA (final conc. 6%), and monopotassium phosphate ($KH_2PO_4$, final conc. 1 mM) were added to the rest of the digested lysate. Lysates were shaken, then spun down for 3 min at 2,000$g$, and supernatants were incubated with $TiO_2$ beads for 5 min at 40°C (1:10, protein/beads, w/w). Beads were washed five times with isopropanol and 5% TFA, and phosphopeptides were eluted off

the beads with 40% acetonitrile (ACN) and 15% of ammonium hydroxide (25% $NH_4OH$) on C8 stage tips. After 20 min of SpeedVac at 45°C, phosphopeptides were desalted on poly(styrenedivinylbenzene)-reversed phase sulfonate (SDB-RPS) stage tips and resolubilized in 5 μl 2% ACN and 0.3% TFA and injected in the mass spectrometer (MS).

### Chromatography and mass spectrometry

Samples were loaded onto 50-cm columns packed in-house with C18 1.9 μM ReproSil particles (Dr. Maisch GmbH), with an EASY-nLC 1000 system (Thermo Fisher Scientific) coupled to the MS (Q Exactive HFX; Thermo Fisher Scientific). A homemade column oven maintained the column temperature at 60°C. Phosphopeptides were eluted with a 140 min gradient starting at 5% buffer B (80% ACN, 0.1% Formic acid) followed by a stepwise increase to 20% in 85 min, 40% in 35 min, 65% in 10 min and 80% in 2 × 5 min at a flow rate of 300 nl/min. Samples were measured in data-dependent acquisition with a (TopN) MS method in which one full scan (300–1,650 m/z, R = 60,000 at 200 m/z, maximum injection time 120 ms) at a target of $3 × 10^6$ ions was first performed, followed by 10 data-dependent MS/MS scans with higher energy collisional dissociation (AGC target $10^5$ ions, maximum injection time at 120 ms, isolation window 1.6 m/z, normalized collision energy 27%, R = 15,000 at 200 m/z). Dynamic exclusion of 40 s and the Apex trigger from 4 to 7 s was enabled.

### Quantification and statistical analysis

MS raw files were processed by the MaxQuant version 1.5.38 (Cox & Mann, 2008) and fragments lists were searched against the mouse uniport FASTA databases (22,220 entries, 39,693 entries, 2015) with cysteine carbamidomethylation as a fixed modification and N-terminal acetylation, methionine oxidations and Serine/Threonine/Tyrosine phosphorylation as variable modifications. We set the FDR to less than 1% at the peptide and protein levels and specified a minimum length of seven amino acids for peptides. Enzyme specificity was set as C-terminal to arginine and lysine as expected using trypsin and lysC as proteases and a maximum of two missed cleavages.

All bioinformatics analyses were performed with the Perseus software (version 1.5.3.0) (Tyanova et al, 2016). Summed intensities were $log_2$-transformed. Samples that did not meet the measurement quality of the overall experiment were excluded. Quantified proteins were filtered for at least 100% of valid values among three or four biological replicates in at least one condition. Missing values were imputed and significantly up- or down-regulated proteins were determined by multiple-sample test (one-way ANOVA, FDR = 0.05). The MS-based proteomics data have been deposited to the ProteomeXchange Consortium via the PRIDE partner repository and are available via ProteomeXchange with identifier PXD028667 (Jones et al, 2008).

### AP-1 transcription factor assay kit

Wild-type BMDMs were stimulated with IL4+IL13+TNF for 15 min (Fig S7F) or with IL4+IL13 for 12 h before TNF was added for further 150 min (Fig S7G). Nuclear extracts were prepared with the NE-PER Nuclear and Cytoplasmic kit (78833; Thermo Fisher Scientific). AP-1 transcription factor assay kit (ab207196; Abcam) was used according to the manufacturer's instructions.

### Quantification and statistical analysis

The significance of differences in the experimental data were determined using GraphPadPrism software. All data involving statistics are presented as mean ± SEM. The number of replicates and the statistical test used are described in the figure legends.

# Data Availability

Data from GSE169348, GSE185380, and PXD028667 were used.

# Supplementary Information

# Acknowledgements

We thank Rin Ho Kim, Marja Driessen, and Assa Yeroslaviz for RNA sequencing and bioinformatics analysis of the RNAseq data; We thank Nina Pijahn for experimental assistance, Katarzyna Grzes for helping with the ATACseq, and Roland Lang for critical appraisal. This work was supported by the Deutsche Forschungsgemeinschaft FOR 2599 (Type 2 Tissue Immunity) (A Roers, SA Eming, EJ Pearce and PJ Murray), the Max Planck Gesellschaft and an Alexander von Humboldt Fellowship (S Dichtl).

## Author Contributions

S Dichtl: conceptualization, data curation, formal analysis, funding acquisition, project administration, and writing—original draft, review, and editing.
DE Sanin: data curation, software, formal analysis, visualization, methodology, and writing—review and editing.
CK Koss: data curation.
S Willenborg: data curation, formal analysis, and methodology.
A Petzold: software, formal analysis, and methodology.
MC Tanzer: data curation, formal analysis, and methodology.
A Dahl: resources and data curation.
AM Kabat: data curation and formal analysis.
L Lindenthal: data curation and methodology.
L Zeitler: data curation and methodology.
S Satzinger: data curation and methodology.
A Strasser: data curation and methodology.
M Mann: resources.
A Roers: resources.
SA Eming: resources.
KC El Kasmi: resources.
EJ Pearce: resources, investigation, and methodology.

P Murray: conceptualization, resources, data curation, supervision, funding acquisition, investigation, methodology, project administration, and writing—original draft, review, and editing.

## Conflict of Interest Statement

No conflicts of interest, financial or otherwise, are declared by the authors. CK Koss and KC El Kasmi are employees of Boehringer Ingelheim Pharma GmbH & Co KG.

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
