## [Reviewer comments · Life Science Alliance]

Life Science Alliance

Gene-selective transcription promotes the inhibition of tissue reparative macrophages by TNF

Stefanie Dichtl, David Sanin, Carolin Koss, Sebastian Willenborg, Andreas Petzold, Maria Tanzer, Andreas Dahl, Agnieszka Kabat, Laura Lindenthal, Leonie Zeitler, Sabrina Satzinger, Alexander Strasser, Matthias Mann, Axel Roers, Sabine Eming, Karim El Kasmi, Edward Pearce, and Peter Murray

DOI: <https://doi.org/10.26508/lsa.202101315>

Corresponding author(s): Peter Murray, Max Planck Institute of Biochemistry

Review Timeline:

Submission Date:	2021-11-24
Editorial Decision:	2021-11-24
Revision Received:	2021-12-22
Editorial Decision:	2021-12-28
Revision Received:	2022-01-03
Accepted:	2022-01-04

Transaction Report:

Please note that the manuscript was previously reviewed at another journal and the reports were taken into account in the decision-making process at *Life Science Alliance*.

1st Review Round

Reviewer #1 Review

Report for Author:
Summary

This study was designed to examine the effect of TNF on IL4/13-stimulated macrophages. The authors conclude that TNF regulates the expression of a subset of M2 polarization-related genes and that this is largely accounted for by JNK signaling. A strength of the study is that anti-TNF therapies are well established in humans; the data presented in this mouse study of TNF treated M2 cells may have mechanistic relevance.

General remarks

This is an interesting manuscript that combines bulk RNA-seq, scRNA-seq, ATAC-seq, and proteomic analysis. One weakness of the study is that these techniques are examined separately with little attempt to employ an integrated systems approach. A second weakness is that the major conclusions presented regarding the role of AP-1 and JNK are not supported by rigorous experimental testing.

Major

1) The RNA-seq data show that TNF selectively blocks the expression of a subset of M2 genes. These data are convincing.

However, the subsequent analysis is limited in scope.

2) The scRNA-seq data are interesting because of the detected heterogeneity. However, no functional relationships between the clusters was examined. Moreover, while the authors conclude that JNK/AP-1 mediate the effects of TNF, no scRNA-seq analysis of macrophages with experimental perturbation of JNK/AP-1 is presented. It is also unclear which sub-cluster includes which TNF-regulated gene expression changes - the limited analysis presented in Fig. 3C should be expanded and quantitated

3) The conclusion that bZip proteins may be relevant to TNF signaling is based on ATAC-seq data is consistent with prior work on bZip proteins and macrophages (cited Ref Focesca et al. 2019). However, there is a disconnect between the limited number of genes regulated by TNF and the large number of ATAC-seq changes. The study would be improved by further comparative analysis of the ATAC-seq data with the RNA-seq data to identify relevant ATAC-seq changes with TNF-regulated gene expression. Moreover, since the authors conclude that JNK/AP1 is relevant, are these changes in ATAC-seq altered by experimental perturbation of JNK/AP-1?

4) The authors conclude that MAPK provides a mechanism for TNF signaling. The conclusion presented is that this MAPK is JNK. It is not clear why other MAPK (e.g. p38 & ERK) are not examined. The key data to test the role of JNK is based on the use of drugs. The conclusions drawn are therefore limited by the specificity of the drugs that are used. The finding that one drug is ineffective (SP600125) and that the other (JNK-in-8) is toxic to macrophages does not lead to confidence in the conclusions that are drawn.

Minor

1) The manuscript would be improved if the authors presented more context for their conclusions regarding the role of JNK in M2 macrophage polarization and function beyond the cited Refs Han et al. (2016 & 2016). For example, Guo et al. (2019) reported that tissue homeostasis regulation by M2 macrophages may be promoted by JNK activation on phagosomes (PMID:31028084).

2) Line 472/3. The authors state that Han et al (2016) showed that phosphorylated Jun was detected in macrophages of JNK1/2 LysM-Cre mice. However, Han et al (2016) presented no data on Jun phosphorylation.

Reviewer #2 Review

Report for Author:

The manuscript by Dichtl et al integrates analysis of several different multi-omics data sets in order to explore the mechanism of how TNF regulates tissue-reparative macrophages. Specifically, the experiments compare BMDMs stimulated with IL-4+IL-13 (M2) with IL-4+IL-13 and TNF co-stimulation. Using RNA-seq they show that TNF negative regulation of IL-4/IL-13 is restricted to a specific subset of genes, and scRNA-seq analysis further showed that only a subset of macrophages were affected by this negative regulation. ATAC-seq analysis showed that TNF reversed changes in chromatin accessibility that were induced by IL-4/IL-13, consistent with reduced expression of a subset of genes. Phospho-proteomics and analysis of the TFs enriched in response to TNF stimulation both pointed to the JNK signaling pathway and JunB as the mediators of the negative regulation, with JNK inhibitors able to reverse the effect.

Overall, the manuscript presents a careful and thorough analysis demonstrating the context-specific role of TNF in modulating M2 macrophages. By using multi-omics approaches, the paper provides a more systematic view of M2 polarization as a set of gene expression modules that are subject to different regulatory pathways—here, TNF negatively regulates some aspects of M2 polarization but not all (for example, they show metabolism is unaffected). This work will be of interest to a wide range of scientists studying macrophage heterogeneity in many different contexts. Below are a few points for the authors to consider:

Major points

1. The authors often refer to genes that are strongly associated with the M2 response. For example, in Fig. 2C-D the authors state that the genes negatively regulated by TNF in the Type 1 cluster are "hallmark" M2 genes (lines 198-200). In Fig. 3, the authors mention that clusters 1 and 2 contain most of the M2 "signature" genes. This raises several questions:

- To what extent do those two groups of gene overlap? It would be helpful for the authors to define which genes they consider to be "M2 signature" genes and then report the extent to which these overlap with the list of genes negatively regulated by TNF.
- To help interpret the scRNA-seq data, it could be informative to visualize an "activity score" for those M2 signature genes (or alternatively for the Type 1 cluster identified in Fig. 2) and look at how they vary together in the scRNA-seq clusters (rather than non-systematically picking 3 genes as in Fig. 3C).

2. Fig. 1 focuses on *Retnla* as the "standard" M2 gene that is negatively regulated by TNF. The authors show consistency of this negative regulation across several contexts. *Arg1* is also reported to act like *Retnla* in some cases, but not all. In Fig. 2, *Arg1* is included in the "Type 1" cluster of genes that are inhibited by TNF (although this appears to be time-dependent). This raises the question of how consistent this Type 1 cluster is across the contexts shown in Fig. 1? Can the authors verify another gene from the Type 1 cluster that shows similar behavior to *Retnla* across the different contexts in Fig. 1?

Minor points

3. In Fig. 3, the authors state that when comparing IL4/IL13 with IL4/IL13+TNF, the scRNA-seq results showed 9 distinct sub-populations and further pinpoint clusters 1,2 and 6 reduced with TNF stim and cluster 0 increased. However, clusters 5 and 8

also appear to be increased with TNF stim. If the graph in Fig. 3A was inverted to show the % of each condition by cluster, it would be possible to show more quantitatively which clusters are enriched for one treatment vs. another.

4. In Fig. 5, the authors suggest that a range of TNFR1-dependent inflammatory pathways can inhibit M2 signaling based on data that IL-1b and IL-36 act similarly to TNF w.r.t. inhibition of IL4/IL13-stimulated *Retnla*. Then in Fig. 6, the authors nicely show dependence on the JNK pathway. Demonstrating dependence on the JNK/Jun for at least one of the other inflammatory pathways would better tie these two figures together. However, I raise it as a minor point because I do not think it's necessary, given that the focus of the paper is TNF.

Reviewer #3 Review

Report for Author:

This study looks at the intersection between signaling by TNF and IL4/IL13. The study is based upon extensive use of anti-TNF directed agents in inflammatory disease and the idea that TNF may contribute to regulation of macrophage polarization. The current title makes no sense. The title needs to include the species and more accurately reflect the actual content of the study e.g. "TNF selectively inhibits the induction of specific target genes in the response of mouse bone marrow macrophages to IL4..." . But in overview, I found it very difficult to see the take home-message and the connection to human disease biology is unclear.

Specific comments

In 2014, the senior author of this work was first author on a series of recommendations regarding macrophage polarization (PMID:25035950). This manuscript largely ignores those recommendations. Few in the field of macrophage biology believe that the M1/M2 concept has any validity or utility. Furthermore, the set of genes that is regulated by IL4R signaling is very poorly conserved in humans (PMID:23293084). None of the genes highlighted in this study is also induced by IL4 in humans. *Retnla* does not even exist in the human genome. Furthermore, the Glass group has recently shown that the response to IL4 is highly divergent even amongst mouse strains (PMID: 34134993).

Against that background, I would suggest that the authors remove all mention of M2 from the abstract and the text and refer to IL4-stimulation/IL4R target genes exclusively. They need to explicitly mention the species and recognize the limitations of studying mouse.

I would note also that the Methods are inadequate with respect to mouse strain. There are critical variants within the C57BL/6 line (e.g. C57Bl/6J has a mutation in the *Nnt* gene that is directly relevant to metabolic function and polarization; and macs in C57Bl/6 also selectively metabolise arginine via *SLC7A2*). So, the authors need to specify that they have worked with the C57BL/6J strain and also how long it has been maintained locally. They also need to provide assurance that any knockouts they analyse are on the same C57BL/6J background.

The recommendations with respect to nomenclature (PMID:25035950) also specifically note the role of CSF1 and confusion surrounding whether CSF1 is an "M2" stimulus. So, the authors do need to recognize they are actually dealing with CSF1 + IL4/13 as the stimulus and the response they observe depends upon both signals. The methods are not adequate in description of exactly how BMDM were made and the source reference quoted also lacks detail. Were they made using CSF1 or L cell conditioned medium (presumably the former, since no source of CSF1 is cited). Cultured for how long? On what substratum? How harvested? Was CSF1 re-added to the medium for subsequent stimulation with TNF or IL4? If so, what concentration? If not, how long after harvest were they restimulated with IL4?

The first sentence of results states that a combination of IL4 + IL13 maximally stimulates IL4RA signaling? Why? This is not explained. Since the IL4 and IL13 receptors have a common signaling pathway, why would the two agonists have an increased effect (and in methods IL4 produced in insect cells (presumably the mouse protein) is not an adequate description of the agonist).

Figure 1 shows the induction of *Retnla* by IL4/13 and the prevention of this response by TNF. The legend does not say whether TNF was added at the same time as IL4/13. In the text, the authors do not actually comment on the most interesting part of this Figure, namely that *RETNLA* protein (is there a reason not to use the gene name for the protein rather than *RELMA*) was detected in only 20% of the stimulated cells (Figure 1C/D). Figure S1 shows similar heterogeneity for induction of *ARG1* (protein name should be capitalized). Also shown in CD301, which is *CLEC10A*, which is said to be an M2 target (on what basis?). In this case, the unstimulated cells are not shown. But taken at face value, Figure S1B would indicate that TNF acts to repress *CLEC10A* largely in those cells that also express *RETNLA*. Given the vagaries of intracellular staining on a flow cytometer, I think it would be desirable to show IF images of these cells stained for both markers.

Figure 2 and Figure S2 relate to the induction of gene expression by IL4/13, TNF and the combination. To evaluate and utilize these data, it would be useful and straightforward to provide an expression Table in Excel format as a supplementary. Heat maps with idiosyncratic choices of genes indicated as in Figure S2 are not useful. One notable inclusion in panel D is *Plau*, which is a CSF1R target gene that rapidly decays when CSF1 is removed and is not further induced by IL4 in CSF1-replete cells (see for example, much earlier data on the response of BMDM to IL4 (PMID: 15908341) which could be compared to results herein). The presence of *Plau* amongst the inducible genes suggests that the cells were actually deprived of CSF1. In the absence of CSF1, TNF (like LPS) can provide a pro-survival signal. Since CSF1 is itself a JNK agonist (as shown in Himes et al.), this key variable is related to subsequent analysis of the role of JNK in the TNF response.

The text related to the RNA-seq data concludes that "One striking aspect of the overall negative effect of TNF is that many of the targeted mRNAs are implicated in hallmark M2 tissue repair and resolution functions such as matrix remodeling (*Retnla*,

Fn1, Angptl2), immune regulation (Arg1, Ccl2), phagocytosis (Mrc1 also known as CD206 or the mannose receptor, Mertk, CD300lf) and cell adhesion (Pecam1, CD34); this pattern is consistent with the original observations about TNF restricting M2 macrophages". This belongs in discussion, the connections to function are tenuous, and it is impossible to interpret without knowing the absolute levels of expression. I strongly doubt that macrophages express PECAM1 or CD34 mRNA or protein at significant levels; and Mrc1 mRNA is highly-expressed by BMDM grown in CSF1 without IL4 stimulation.

Figure 3 describes single cell RNA-seq analysis. The data in Figure 1 already showed that only a subset of cells responded to IL4/IL13 with induction of Retnla but this work fails to provide or even discuss a mechanism. The rationale for this further analysis is not clear and there are few obvious biologies associated with the different clusters in Figure S3, except that Cluster 3 is obviously associated with the cell cycle, and Cluster 4 is associated with Class II MHC expression. Populations 1 and 2 are said to be enriched for the M2 signature. In what way? Figure 3C shows only 3 genes, and two of them (Mgl2 and Clec10A) do not actually show any evident enrichment in Clusters 1 and 2. Population 0 is said to have "a dominant gene expression signature of inflammatory-associated gene expression". Cannot see any evidence of this from the genes shown in Figure S3A; TNF alone (Figure S3B) does not demonstrate such a claim. In my view, the separation of Clusters 0, 1 and 2 is likely an artefact of scRNA-seq and over-analysis that actually reflects quite small relative differences in gene expression. It certainly requires further validation if these genes are supposedly indicative of genuine heterogeneity amongst BMDM.

Figure 4 describes transcriptional regulation of target genes by IL4/13 and demonstrates that TNF does inhibit transcription initiation and has an impact on local chromatin structure (4D). Legend refers to significance without indicating the number of replicates or the basis of quantitation. The interpretation of ATAC-seq data is complex; is the increased signal due to increased access in individual cells or more cells with open chromatin.

Figure S5 deal with the effect of TNF and IL4/13 on metabolism. Since nitric oxide metabolism is a rodent-specific biology in macrophages, and also uniquely prevalent even within mice strains to C57BL/6J, the relevance to humans is not clear. TNF is said to induce many genes associated with pro-inflammatory metabolism, but the data actually show that the induction is minimal compared to the response to LPS. I did not feel that this section made any contribution to the paper.

Figure 5 makes two simple points; that the reduced induction of Retnla by IL4/13 is also seen with IL1 or IL36, and the response to TNF depends upon TNF receptor TNFR1. Do the TNFR1 ^{-/-} macrophages respond to IL1?

The final section starts with the statement "To evaluate transcription factors potentially regulated by TNF in the context of IL4+IL13 signaling, we performed a phospho-proteomic study". How does this follow? It is not at all clear what question is being asked. The following section (pp10-12) is much too long and contains a great deal of discussion that is not clearly to the content of Figure 6 and the supplementary Table. One interesting and likely crucial observation is the loss of CSF1R-associated phosphopeptides, since CSF1R is subject to ectodomain cleavage by Adam17. Otherwise, given the literature, there is an obvious cut to the chase that TNF is known to activate the JNK pathway and increase nuclear AP1. That said, the effects of JNK inhibition are at best pharmacological and give no new insight into TNF action. Jun is phosphorylated constitutively in BMDM in the presence of CSF1. The cited paper from Himes et al. showed changes in gene expression in BMDM within 2 hours of JNK inhibition. So, the inhibitors likely act in part by blocking CSF1R signals.

November 24, 2021

Re: Life Science Alliance manuscript #LSA-2021-01315-T

Peter Murray
Max Planck Institute of Biochemistry

Dear Dr. Murray,

Thank you for submitting your manuscript entitled "Gene-selective transcription promotes the inhibition of tissue reparative macrophages by TNF" to Life Science Alliance. We invite you to submit a revised manuscript addressing the following:

- Address Reviewer 1's comments. You can either include experiments perturbing JNK signaling, or present the manuscript more as a Resource and tone down the JNK conclusions.
- Address Reviewer 2's comments.
- Address Reviewer 3's comments, without the need to remove the mention or focus on M2.

Thank you for this interesting contribution to Life Science Alliance. We are looking forward to receiving your revised manuscript.

Sincerely,

- A letter addressing the reviewers' comments point by point.
- An editable version of the final text (.DOC or .DOCX) is needed for copyediting (no PDFs).
- High-resolution figure, supplementary figure and video files uploaded as individual files: See our detailed guidelines for preparing your production-ready images, <https://www.life-science-alliance.org/authors>
- Summary blurb (enter in submission system): A short text summarizing in a single sentence the study (max. 200 characters including spaces). This text is used in conjunction with the titles of papers, hence should be informative and complementary to the title and running title. It should describe the context and significance of the findings for a general readership; it should be written in the present tense and refer to the work in the third person. Author names should not be mentioned.
- By submitting a revision, you attest that you are aware of our payment policies found here: <https://www.life-science-alliance.org/copyright-license-fee>

B. MANUSCRIPT ORGANIZATION AND FORMATTING:

Revision Overview:

Originally, this manuscript was submitted to *another journal* and received a detailed review. Following the rejection notice, we transferred the manuscript to LSA. Dr. Sawey provided an overview to improve the manuscript. These comments were:

- Address Reviewer 1's comments. You can either include experiments perturbing JNK signaling, or present the manuscript more as a Resource and tone down the JNK conclusions.
- Address Reviewer 2's comments.
- Address Reviewer 3's comments, without the need to remove the mention or focus on M2.

We have completed these revisions, which are incorporated into the comments to each reviewer below. The reviewer comments are copied directly from the decision email. Specific comments are in **red**. In addition, the changes to the manuscript text are also in red, including the new title.

Reviewer #1:

Summary

This study was designed to examine the effect of TNF on IL4/13-stimulated macrophages. The authors conclude that TNF regulates the expression of a subset of M2 polarization-related genes and that this is largely accounted for by JNK signaling. A strength of the study is that anti-TNF therapies are well established in humans; the data presented in this mouse study of TNF treated M2 cells may have mechanistic relevance.

General remarks

This is an interesting manuscript that combines bulk RNA-seq, scRNA-seq, ATAC-seq, and proteomic analysis. One weakness of the study is that these techniques are examined separately with little attempt to employ an integrated systems approach. A second weakness is that the major conclusions presented regarding the role of AP-1 and JNK are not supported by rigorous experimental testing.

Major

1) The RNA-seq data show that TNF selectively blocks the expression of a subset of M2 genes. These data are convincing. However, the subsequent analysis is limited in scope.

This remark is general and, therefore, we see this point more as an opinion of the reviewer.

2) The scRNA-seq data are interesting because of the detected heterogeneity. However, no functional relationships between the clusters was examined. Moreover, while the authors conclude that JNK/AP-1 mediate the effects of TNF, no scRNA-seq analysis of macrophages with experimental perturbation of JNK/AP-1 is presented. It is also unclear which sub-cluster includes which TNF-regulated gene expression changes - the limited analysis presented in Fig. 3C should be expanded and quantitated.

The suggestion to add a scRNAseq analysis in the presence of JNK-inhibitors is beyond the scope of this manuscript. From the Editorial comment, we modified the JNK conclusions throughout the manuscript. We added further analysis of the scRNAseq data (which was also suggested by reviewer 2). Additionally, we expanded the quantification of the scRNAseq data, which is now provided in Supplementary Figure 4 (Panels B, D, E).

3) The conclusion that bZip proteins may be relevant to TNF signaling is based on ATAC-seq data is consistent with prior work on bZip proteins and macrophages (cited Ref Focesca et al. 2019). However, there is a disconnect between the limited number of genes regulated by TNF

and the large number of ATAC-seq changes. The study would be improved by further comparative analysis of the ATAC-seq data with the RNA-seq data to identify relevant ATAC-seq changes with TNF-regulated gene expression. Moreover, since the authors conclude that JNK/AP1 is relevant, are these changes in ATAC-seq altered by experimental perturbation of JNK/AP-1?

The reviewer asks about the connection between the number of ATACseq changes versus the number of genes regulated by TNF. First, we focused the manuscript on the IL4+IL13 mRNAs negatively regulated by TNF. This is the first time such an analysis has been done at this granularity and certainly, much more work will be needed to understand the complexity of the interplay between these cytokine signaling pathways. Second, to add ATACseq data using experimental perturbation of JNK/AP-1 is far beyond the scope of this manuscript (although we certainly view this as worthy). Thus, the more detailed analysis of all the relevant B-ZIP proteins in conjunction with genetic and pharmacological perturbation of JNK signaling is an entirely new project. Third, we provided such an integrative analysis in the volcano plots in Fig. 4. There we highlight the peaks in the ATACseq data that match some of the genes we are interested in based on the RNAseq. The ATAC-seq experiment is just one small part of the manuscript that supplied a piece of the puzzle.

4) The authors conclude that MAPK provides a mechanism for TNF signaling. The conclusion presented is that this MAPK is JNK. It is not clear why other MAPK (e.g. p38 & ERK) are not examined. The key data to test the role of JNK is based on the use of drugs. The conclusions drawn are therefore limited by the specificity of the drugs that are used. The finding that one drug is ineffective (SP600125) and that the other (JNK-in-8) is toxic to macrophages does not lead to confidence in the conclusions that are drawn.

SP600125 is not “ineffective” as it is a (ATP) competitive inhibitor of JNKs, while the JNK-IN-X series are irreversibly inhibitors. This difference is carefully articulated in the manuscript. In addition, JNK-IN-8 is toxic during macrophage development under the influence of CSF1 signaling, consistent with Hume’s data. The fact that the inhibitors work differently produced the predicted results we obtained: JNK-IN-8 is a potent JNK blocker.

To answer the first part of the comment, we used inhibitors for the ERK pathway (Trametinib), and for p38 (PH-797804) and found that these inhibitors were not able to inhibit Arg1 mRNA expression compared to the effect of JNK-IN-8 (Bar graph below). The reviewer should note that previous work by our group and Sid Morris (e.g. Pauleau et al. J. Immunol. And El Kasmi et al. Nature Immunol.) showed C/EBP β as a key B-ZIP factor in Arg1 regulation, which is regulated in part by p38/ERK. The phosphoproteomics gave a clue that JunB/JunD was the most likely candidate linked to the effects of TNF in the context of IL4+IL13 signaling. Therefore, we further analyzed the phosphorylation of JNK and JunB in the presence of ERK and p38 inhibitors; the presence of the p38 inhibitor didn’t change pJNK nor pJunB (Western blot below). In contrast, Trametinib, which reduced phosphorylation of JunB but not JNK. In order to establish the relationship between the MAPK pathways and the downstream gene expression effects, a comparative study is needed where on-target drugs (Trametinib, JNK-IN-8) are used in a side-by-side way. Our work, and that of the Glass lab using molecular genetic approaches, suggests such a complex system requires an equivalently complex experimental approach to truly understand how the system works; this is an important future objective.

Minor

1) The manuscript would be improved if the authors presented more context for their conclusions regarding the role of JNK in M2 macrophage polarization and function beyond the cited Refs Han et al. (2016 & 2016). For example, Guo et al. (2019) reported that tissue homeostasis regulation by M2 macrophages may be promoted by JNK activation on phagosomes (PMID:31028084).

We agree and we added this reference as suggested.

2) Line 472/3. The authors state that Han et al (2016) showed that phosphorylated Jun was detected in macrophages of JNK1/2 LysM-Cre mice. However, Han et al (2016) presented no data on Jun phosphorylation.

Han et al. showed p-Jun data in their Supplemental file (Fig. S3), which is directly copied from their paper below. Note that their western data in panel A shows an incomplete effect of the knockout reading out JNK (which may still produce a strong biological outcome in some settings). Their flow data for p-Jun Ser63 is difficult to understand but shows a partial effect (comparing the relative position of the blue lines).

Figure S3. Characterization of JNK-deficient macrophages.

Fig. S3

(A) Bone marrow-derived macrophages (BMDM) isolated from Φ^{WT} and Φ^{KO} mice were examined by immunoblot analysis by probing with antibodies to JNK1/2 and α Tubulin.

(B) BMDM were treated without (Control) and with 10 ng/ml LPS (30 min). The cells were examined by immunofluorescence microscopy by staining with antibodies to pSer⁶³-cJun (green) and α Tubulin (red). Scale bar, 100 μ m.

(C) BMDM were stained with an antibody to pSer⁶³-cJun and examined by flow cytometry.

Reviewer #2:

The manuscript by Dichtl et al integrates analysis of several different multi-omics data sets in order to explore the mechanism of how TNF regulates tissue-reparative macrophages. Specifically, the experiments compare BMDMs stimulated with IL-4+IL-13 (M2) with IL-4+IL-13 and TNF co-stimulation. Using RNA-seq they show that TNF negative regulation of IL-4/IL-13 is restricted to a specific subset of genes, and scRNA-seq analysis further showed that only a subset of macrophages were affected by this negative regulation. ATAC-seq analysis showed that TNF reversed changes in chromatin accessibility that were induced by IL-4/IL-13, consistent with reduced expression of a subset of genes. Phospho-proteomics and analysis of the TFs enriched in response to TNF stimulation both pointed to the JNK signaling pathway and JunB as the mediators of the negative regulation, with JNK inhibitors able to reverse the effect.

Overall, the manuscript presents a careful and thorough analysis demonstrating the context-specific role of TNF in modulating M2 macrophages. By using multi-omics approaches, the paper provides a more systematic view of M2 polarization as a set of gene expression modules that are subject to different regulatory pathways—here, TNF negatively regulates some aspects of M2 polarization but not all (for example, they show metabolism is unaffected). This work will be of interest to a wide range of scientists studying macrophage heterogeneity in many different contexts. Below are a few points for the authors to consider:

We thank this reviewer for the detailed appraisal, which detected some errors and inconsistencies we were able to clarify and resolve below.

Major points

1. The authors often refer to genes that are strongly associated with the M2 response. For example, in Fig. 2C-D the authors state that the genes negatively regulated by TNF in the Type 1 cluster are "hallmark" M2 genes (lines 198-200). In Fig. 3, the authors mention that clusters 1 and 2 contain most of the M2 "signature" genes. This raises several questions:

a. To what extent do those two groups of gene overlap? It would be helpful for the authors to define which genes they consider to be "M2 signature" genes and then report the extent to which these overlap with the list of genes negatively regulated by TNF.

We analyzed the type 1 and 2 genes in the scRNAseq data and were able to find the following mRNAs: *Retnla*, *Nes*, *Mamdc2*, *Gypc*, *Kitl*, *Proz*, *Chst11* and *Mrc1*, which are type 1 or 2 mRNAs and fall in cluster 1 & 2 in the scRNAseq data. We added this list of genes to the manuscript and included the figure in supplemental Fig.4 and further modified the text in this regard.

b. To help interpret the scRNA-seq data, it could be informative to visualize an "activity score" for those M2 signature genes (or alternatively for the Type 1 cluster identified in Fig. 2) and look at how they vary together in the scRNA-seq clusters (rather than non-systematically picking 3 genes as in Fig. 3C).

We included the graphs visualizing "activity scores" of M2 and Type 1 and 2 genes in Supplemental Fig.4.

2. Fig. 1 focuses on *Retnla* as the "standard" M2 gene that is negatively regulated by TNF. The authors show consistency of this negative regulation across several contexts. *Arg1* is also reported to act like *Retnla* in some cases, but not all. In Fig. 2, *Arg1* is included in the "Type 1" cluster of genes that are inhibited by TNF (although this appears to be time-dependent). This raises the question of how consistent this Type 1 cluster is across the contexts shown in Fig. 1? Can the authors verify another gene from the Type 1 cluster that shows similar behavior to *Retnla* across the different contexts in Fig. 1?

Another gene from the Type 1 cluster *Mrc1* (encodes CD206). We show that CD206 expression is also decreased with the addition of TNF. Further the *Mrc1* expression displays the same pattern as *Retnla* in the *in vivo* wound healing model and is significantly increased in

Etanercept-treated mice, where SPMs were isolated. These data are shown in Fig. S2.

Minor points

3. In Fig. 3, the authors state that when comparing IL4/IL13 with IL4/IL13+TNF, the scRNA-seq results showed 9 distinct sub-populations and further pinpoint clusters 1,2 and 6 reduced with TNF stim and cluster 0 increased. However, clusters 5 and 8 also appear to be increased with TNF stim.

We agree. We changed the text in the manuscript relevant to this point.

If the graph in Fig. 3A was inverted to show the % of each condition by cluster, it would be possible to show more quantitatively which clusters are enriched for one treatment vs. another.

We added this quantification in Supplemental Fig. 4B.

4. In Fig. 5, the authors suggest that a range of TNFR1-dependent inflammatory pathways can inhibit M2 signaling based on data that IL-1b and IL-36 act similarly to TNF w.r.t. inhibition of IL4/IL13-stimulated *Retnla*. Then in Fig. 6, the authors nicely show dependence on the JNK pathway. Demonstrating dependence on the JNK/Jun for at least one of the other inflammatory pathways would better tie these two figures together. However, I raise it as a minor point because I do not think it's necessary, given that the focus of the paper is TNF.

We included this experiment in Supplemental Fig 7E. The combination of IL4+IL13+IL1 β also induced p-JunB in wild type BMDMs and was impaired in *Tnfr1*^{-/-} BMDMs.

Reviewer #3:

This study looks at the intersection between signaling by TNF and IL4/IL13. The study is based upon extensive use of anti-TNF directed agents in inflammatory disease and the idea that TNF may contribute to regulation of macrophage polarization. The current title makes no sense. The title needs to include the species and more accurately reflect the actual content of the study e.g. "TNF selectively inhibits the induction of specific target genes in the response of mouse bone marrow macrophages to IL4..." . But in overview, I found it very difficult to see the take home message and the connection to human disease biology is unclear.

Specific comments

In 2014, the senior author of this work was first author on a series of recommendations regarding macrophage polarization (PMID:25035950). This manuscript largely ignores those recommendations. Few in the field of macrophage biology believe that the M1/M2 concept has any validity or utility. Furthermore, the set of genes that is regulated by IL4R signaling is very poorly conserved in humans (PMID:23293084). None of the genes highlighted in this study is also induced by IL4 in humans. *Retnla* does not even exist in the human genome. Furthermore, the Glass group has recently shown that the response to IL4 is highly divergent even amongst mouse strains (PMID: 34134993).

Against that background, I would suggest that the authors remove all mention of M2 from the abstract and the text and refer to IL4-stimulation/IL4R target genes exclusively. We cover these more philosophical points explicitly in the Introduction, noting the Editorial comments in this regard.

- They need to explicitly mention the species and recognize the limitations of studying mouse.
I would note also that the Methods are inadequate with respect to mouse strain. There are critical variants within the C57BL/6 line (e.g. C57Bl/6J has a mutation in the *Nnt* gene that is directly relevant to metabolic function and polarization; and macs in C57Bl/6 also selectively metabolise arginine via *SLC7A2*). So, the authors need to specify that they have worked with the C57BL/6J strain and also how long it has been

maintained locally. They also need to provide assurance that any knockouts they analyse are on the same C57BL/6J background.

We specified mouse background in the materials part of the manuscript.

- The recommendations with respect to nomenclature (PMID:25035950) also specifically note the role of CSF1 and confusion surrounding whether CSF1 is an "M2" stimulus. So, the authors do need to recognize they are actually dealing with CSF1 + IL4/13 as the stimulus and the response they observe depends upon both signals. The methods are not adequate in description of exactly how BMDM were made and the source reference quoted also lacks detail. Were they made using CSF1 or L cell conditioned medium (presumably the former, since no source of CSF1 is cited). Cultured for how long? On what substratum? How harvested? Was CSF1 re-added to the medium for subsequent stimulation with TNF or IL4? If so, what concentration? If not, how long after harvest were they restimulated with IL4?

The first sentence of results states that a combination of IL4 + IL13 maximally stimulates IL4RA signaling? Why? This is not explained. Since the IL4 and IL13 receptors have a common signaling pathway, why would the two agonists have an increased effect (and in methods IL4 produced in insect cells (presumably the mouse protein) is not an adequate description of the agonist).

In all studies using BMDMs from our group, we use highly purified (Baculovirus) recombinant human CSF1. We included the missing information articulated above mainly in the Methods section. We do not use LCCM due to our own experience and the proteomic information recently published in Life Science Alliance concerning the many components in the supernatant that could be confounding factors in macrophage growth and activation. An exception is when we use CSF1 in comparison to LCCM to point out methodological issues in other publications (Dichtl et al. 2021, Science Advances, for example). Regarding the last point, we generally use both IL4R-alpha cytokines to obtain "maximal" signal through the pathway. These cytokines are mouse.

- Figure 1 shows the induction of Retnla by IL4/13 and the prevention of this response by TNF. The legend does not say whether TNF was added at the same time as IL4/13. In the text, the authors do not actually comment on the most interesting part of this Figure, namely that RETNLA protein (is there a reason not to use the gene name for the protein rather than RELMa) was detected in only 20% of the stimulated cells (Figure 1C/D). Figure S1 shows similar heterogeneity for induction of ARG1 (protein name should be capitalized). Also shown in CD301, which is CLEC10A, which is said to be an M2 target (on what basis?) In this case, the unstimulated cells are not shown. But taken at face value, Figure S1B would indicate that TNF acts to repress CLEC10A largely in those cells that also express RETNLA. Given the vagaries of intracellular staining on a flow cytometer, I think it would be desirable to show IF images of these cells stained for both markers.

"the authors do not actually comment on the most interesting part of this Figure": the heterogeneity is indeed fascinating, as it suggests on a fraction of the macrophages can respond. This is why we did the scRNAseq analysis, which agrees with recent work from Miller-Jensen's group (cited). However, this concept is "new" to many people in this field, and thus we were careful in describing the outcomes.

We changed the figure legends accordingly and now use RELMa. We further included a more detailed description of the experimental procedure in the Methods and highlight the percentage of RELM α^+ cells in the text. CLEC10A was previously shown to be an "M2" gene (Raes, Brys et al., 2005). Our results substantiate this finding. We added the bar graph displaying that IL4+IL13-stimulated BMDMs were significantly increased compared to untreated cells in Supplemental Fig. 1C.

We disagree with the opinion of this reviewer regarding the vagaries of the intracellular staining on a flow cytometer. We show highly reproducible data which were also

confirmed in other laboratories and is routinely used in the Pearce group. Therefore, we think that the intracellular staining of the shown markers on a flow cytometer is accurate and reflects the biology of the system.

- Figure 2 and Figure S2 relate to the induction of gene expression by IL4/13, TNF and the combination. To evaluate and utilize these data, it would be useful and straightforward to provide an expression Table in Excel format as a supplementary. All data are uploaded as GSE files and can therefore be used by anyone. Heat maps with idiosyncratic choices of genes indicated as in Figure S2 are not useful. Likely the reviewer means the heat maps shown in Figure S2A and B. These heatmaps have the purpose to highlight the distribution of the different groups of regulation over time. To generate these heatmaps we analyzed the 100 most regulated genes of each time point. Therefore, the gene names which are highlighted are from an unbiased analysis and not an idiosyncratic choice. One notable inclusion in panel D is Plau, which is a CSF1R target gene that rapidly decays when CSF1 is removed and is not further induced by IL4 in CSF1-replete cells (see for example, much earlier data on the response of BMDM to IL4 (PMID: 15908341) which could be compared to results herein). The presence of Plau amongst the inducible genes suggests that the cells were actually deprived of CSF1. Figure 2C and D show the unbiased regulation of mRNAs, which were categorized by regulation pattern type. As we note in the Methods, CSF1 was always present during the stimulation. We also checked the scRNAseq data and found that the Plau expression was undetectable, further highlighting the difference between bulk and scRNAseq (this may be technical in nature) In the absence of CSF1, TNF (like LPS) can provide a pro-survival signal. Since CSF1 is itself a JNK agonist (as shown in Himes et al.), this key variable is related to subsequent analysis of the role of JNK in the TNF response. In all experiments CSF1 was added as mentioned in the Methods.
- The text related to the RNA-seq data concludes that "One striking aspect of the overall negative effect of TNF is that many of the targeted mRNAs are implicated in hallmark M2 tissue repair and resolution functions such as matrix remodeling (Retnla, Fn1, Angptl2), immune regulation (Arg1, Ccl2), phagocytosis (Mrc1 also known as CD206 or the mannose receptor, Mertk, CD300lf) and cell adhesion (Pecam1, CD34); this pattern is consistent with the original observations about TNF restricting M2 macrophages". This belongs in discussion, the connections to function are tenuous, and it is impossible to interpret without knowing the absolute levels of expression. I strongly doubt that macrophages express PECAM1 or CD34 mRNA or protein at significant levels; and Mrc1 mRNA is highly-expressed by BMDM grown in CSF1 without IL4 stimulation. We analyzed the expression of the described genes and found that Pecam1 and CD34 were not present in the scRNAseq data. Therefore, we left the heatmaps of the mRNA types as they are because it was an unbiased analysis. This analysis highlights the advantage of scRNAseq compared to bulk RNAseq. Mrc1 expression is further increased with IL4/IL13 stimulation. We used this mRNA to show another example of comparably regulated like Retnla (see point 2 of reviewer 2).
- Figure 3 describes single cell RNA-seq analysis. The data in Figure 1 already showed that only a subset of cells responded to IL4/IL13 with induction of Retnla but this work fails to provide or even discuss a mechanism. The rationale for this further analysis is not clear and there are few obvious biologies associated with the different clusters in Figure S3, except that Cluster 3 is obviously associated with the cell cycle, and Cluster 4 is associated with Class II MHC expression. Populations 1 and 2 are said to be enriched for the M2 signature. In what way? We added further analysis of our scRNAseq data, which show that the cluster 1 & 2 are enriched for M2 and type 1 genes. These additional figures are in Supplemental Fig. 4.

- Figure 3C shows only 3 genes, and two of them (Mgl2 and Clec10A) do not actually show any evident enrichment in Clusters 1 and 2.
We consider that the reviewer misinterpreted the figure. As explained in the figure legend, Fig. 3C only shows IL4/IL13 vs IL4/IL13 + TNF (without an unstimulated control group). Therefore, the figures only show a decreased expression of cluster 1 & 2 in the IL4/IL13 + TNF group.

Population 0 is said to have "a dominant gene expression signature of inflammatory-associated gene expression". Cannot see any evidence of this from the genes shown in Figure S3A; TNF alone (Figure S3B) does not demonstrate such a claim. In my view, the separation of Clusters 0, 1 and 2 is likely an artefact of scRNA-seq and over-analysis that actually reflects quite small relative differences in gene expression. It certainly requires further validation if these genes are supposedly indicative of genuine heterogeneity amongst BMDM.

Population 0 is significantly enriched for mRNAs including Sod2, Clec4e, Cxcl2, Nfkb1a, Cxcl1 and Acod1, which all are "pro-inflammatory" (i.e., induced by LPS stimulation as we discuss in detail). TNF is also significantly upregulated in cluster 0, further substantiating our conclusion.

We demonstrate that the clusters we identified using well documented and nearly universally accepted methodologies for single cell RNA sequencing (see Methods) are transcriptionally distinct. This is indicated in Fig. S3A where each cluster is associated with several statistically significant differentially expressed genes. The fact that each cluster can be associated with a statistically significant and unique transcriptional signature is in and of itself a validation of the clustering strategy. The reviewer refers to these differences as "quite small", yet the number of unique statistically significant differentially expressed genes we identified was 2894 across all the clusters. Moreover, we have indicated in our methods the controls we have put in place to avoid over-clustering of data. Although we document this transcriptional diversity in our manuscript, our focus is on the effects of TNF, so further characterization of these subpopulations is beyond the scope of this manuscript.

- Figure 4 describes transcriptional regulation of target genes by IL4/13 and demonstrates that TNF does inhibit transcription initiation and has an impact on local chromatin structure (4D). Legend refers to significance without indicating the number of replicates or the basis of quantitation. The interpretation of ATAC-seq data is complex; is the increased signal due to increased access in individual cells or more cells with open chromatin.

Increased access in individual cells and more cells with open chromatin indicates the same outcome: increased accessibility in the regions. To go in more detail we would need to perform scATACseq, which as we noted above, is beyond the scope of this already detailed manuscript. We described the analysis of the ATACseq in a very detailed way in the methods. We used triplicates and added this information to the figure legend.

- Figure S5 deal with the effect of TNF and IL4/13 on metabolism. Since nitric oxide metabolism is a rodent-specific biology in macrophages, and also uniquely prevalent even within mice strains to C57BL/6J, the relevance to humans is not clear. TNF is said to induce many genes associated with pro-inflammatory metabolism, but the data actually show that the induction is minimal compared to the response to LPS. I did not feel that this section made any contribution to the paper.

This is an opinion of the reviewer. We consider this part of the manuscript important and relevant for further investigations as also provides a reference point for other investigators working in this area. Therefore, we retained the "metabolic" part as a supplementary figure.

- Figure 5 makes two simple points; that the reduced induction of *Retnla* by IL4/13 is also seen with IL1 or IL36, and the response to TNF depends upon TNF receptor TNFR1. Do the TNFR1 ^{-/-} macrophages respond to IL1?

Tnfr1^{-/-} BMDMs didn't show a significant reduction of *Retnla* expression following IL4/IL13+IL1 β treatment (compared to IL4/IL13), which was the case for WT BMDMs. According to the suggestion of reviewer 2 we also analyzed the pJunB expression of IL4/IL13+IL1 β stimulated WT and *Tnfr1^{-/-}* BMDMs and found that IL4/IL13+IL1 β treatment increases p-JunB expression which was significantly decreased in *Tnfr1^{-/-}* BMDMs (Supplemental Fig. 7E).

- The final section starts with the statement "To evaluate transcription factors potentially regulated by TNF in the context of IL4+IL13 signaling, we performed a phospho-proteomic study". How does this follow? It is not at all clear what question is being asked. The following section (pp10-12) is much too long and contains a great deal of discussion that is not clearly to the content of Figure 6 and the supplementary Table. One interesting and likely crucial observation is the loss of CSF1R-associated phosphopeptides, since CSF1R is subject to ectodomain cleavage by Adam17. Otherwise, given the literature, there is an obvious cut to the chase that TNF is known to activate the JNK pathway and increase nuclear AP1. That said, the effects of JNK inhibition are at best pharmacological and give no new insight into TNF action. Jun is phosphorylated constitutively in BMDM in the presence of CSF1. The cited paper from Himes et al. showed changes in gene expression in BMDM within 2 hours of JNK inhibition. So, the inhibitors likely act in part by blocking CSF1R signals.
 We agree with the reviewer's comment that the inhibitors could act in part by blocking CSF1R signaling. However, due to the limited possibilities to perturb the JNK signaling pathway (i.e. molecular genetic tools, such as the conditional JunB knockouts are available at JAX but frozen, and our colleagues working in this area do not have living colonies of mice, and incorporation of a strain such as this requires permission from the German authorities, which is a process currently taken >6 months, even for revisions to an existing license), we did not straightforward ways to overcome this problem.

Reference:

Raes G, Brys L, Dahal BK, Brandt J, Grooten J, Brombacher F, Vanham G, Noel W, Bogaert P, Boonefaes T, Kindt A, Van den Bergh R, Leenen PJ, De Baetselier P, Ghassabeh GH (2005) Macrophage galactose-type C-type lectins as novel markers for alternatively activated macrophages elicited by parasitic infections and allergic airway inflammation. *J Leukoc Biol* 77: 321-7

December 28, 2021

RE: Life Science Alliance Manuscript #LSA-2021-01315-TR

Prof. Peter Murray
Max Planck Institute of Biochemistry
Am Klopferspitz 18
Martinsried 82152
Germany

Dear Dr. Murray,

Thank you for submitting your revised manuscript entitled "Gene-selective transcription promotes the inhibition of tissue reparative macrophages by TNF". We would be happy to publish your paper in Life Science Alliance pending final revisions necessary to meet our formatting guidelines.

- please upload your main and supplementary figures as single files
- please add ORCID ID for the corresponding author-you should have received instructions on how to do so
- please add the Twitter handle of your host institute/organization as well as your own or/and one of the authors in our system
- tables should be included at the bottom of the main manuscript file or be sent as separate files
- please use the [10 author names, et al.] format in your references (i.e. limit the author names to the first 10)
- please add a Data Availability Statement summarizing the accession numbers for the sequencing data

FIGURE CHECKS:

- Resolution of the figures should be increased to at least 300 dpi
- If the vertical lines placed over the blots in Supplemental Figures 6D and S7E represent splicing to remove irrelevant lanes, please indicate this in the figure legends. If they are meant to indicate something else, please mention why they are there.

A. FINAL FILES:

B. MANUSCRIPT ORGANIZATION AND FORMATTING:

Sincerely,

January 4, 2022

RE: Life Science Alliance Manuscript #LSA-2021-01315-TRR

Prof. Peter Murray
Max Planck Institute of Biochemistry
Am Klopferspitz 18
Martinsried 82152
Germany

Dear Dr. Murray,

Thank you for submitting your Research Article entitled "Gene-selective transcription promotes the inhibition of tissue reparative macrophages by TNF". It is a pleasure to let you know that your manuscript is now accepted for publication in Life Science Alliance. Congratulations on this interesting work.

DISTRIBUTION OF MATERIALS:

Again, congratulations on a very nice paper. I hope you found the review process to be constructive and are pleased with how the manuscript was handled editorially. We look forward to future exciting submissions from your lab.

Sincerely,
